



# Cloud top heights and aerosol columnar properties from combined EarthCARE lidar and imager observations: the AM-CTH and AM-ACD products

Moritz Haarig[1], Anja Hünerbein[1], Ulla Wandinger[1], Nicole Docter[2], Sebastian Bley[1], David Donovan[3], and Gerd-Jan van Zadelhoff[3]

[1]Leibniz Institute for Tropospheric Research (TROPOS), Leipzig, Germany
[2]Free University of Berlin (FUB), Institute for Space Science, Berlin, Germany
[3]Royal Netherlands Meteorological Institute (KNMI), De Bilt, The Netherlands

**Correspondence:** Moritz Haarig (haarig@tropos.de)

**Abstract.** The Earth Clouds, Aerosols and Radiation Explorer (EarthCARE) is the combination of multiple active and passive instruments on a single platform. The Atmospheric Lidar (ATLID) provides vertical information of clouds and aerosol particles along the satellite track. In addition, the Multi-Spectral Imager (MSI) collects the multispectral information from the visible till the infrared wavelengths over a swath width of 150 km across the track. The ATLID–MSI Column Products processor (AM-COL) described in this paper combines the high vertical resolution of the lidar along track and the horizontal resolution of the imager across track to better characterize the 3-dimensional scene. ATLID Level 2a (L2a) data from the ATLID Layer Products processor (A-LAY) and MSI L2a data from the MSI Cloud Products processor (M-CLD) and the MSI Aerosol Optical Thickness processor (M-AOT) as well as MSI Level 1c (L1c) data are used as input to produce the synergistic columnar products: the ATLID–MSI Cloud Top Height (AM-CTH) and the ATLID–MSI Aerosol Column Descriptor (AM-ACD). The coupling of ATLID (measuring at 355 nm) and MSI (at $\geq 670$ nm) provides multispectral observations of the aerosol properties. Especially, the Ångström exponent from the spectral aerosol optical thickness (AOT 355 nm/670 nm) adds valuable information for aerosol typing. The AOT across track, the Ångström exponent and the dominant aerosol type are stored in the AM-ACD product. The accurate detection of the Cloud Top Height (CTH) with lidar is limited to the ATLID track. The difference of the CTH detected by ATLID and MSI is calculated along track. The similarity of MSI pixels across track with those along track is used to transfer the calculated CTH difference to the entire MSI swath. In this way, the accuracy of the CTH is increased to achieve the EarthCARE mission goal aiming to derive the radiative flux at the top of the atmosphere with an accuracy of $10 \text{ Wm}^{-2}$ for a $100 \text{ km}^2$ snapshot view of the atmosphere. The synergistic CTH difference is stored in the AM-CTH product. The quality status depending on day/night conditions or the presence of multiple cloud or aerosol layers is provided with the products. The algorithm was successfully tested using the common EarthCARE test scenes. Two definitions of the CTH from the model-truth cloud extinction fields are compared: An extinction-based threshold of $20 \text{ Mm}^{-1}$ provides the geometric CTH and a cloud-optical-thickness threshold of 0.25 describes the radiative CTH. The first one is detected with ATLID, the second one with MSI.



## 1 Introduction

Clouds and aerosol particles have a major influence on the radiation budget of the Earth as they interact with incoming solar
radiation and outgoing terrestrial radiation. However, their global distribution is highly variable in time and space. Additionally, their vertical distribution is essential to accurately calculate their role in the radiation budget. To improve the global
observation capabilities and the radiation models, the Earth Clouds, Aerosols and Radiation Explorer (EarthCARE) mission
was designed (Illingworth et al., 2015). The European Space Agency (ESA) and the Japan Aerospace Exploration Agency
(JAXA) built a satellite with four instruments on one single platform: a Cloud-Profiling Radar (CPR), an Atmospheric Lidar
(ATLID), a Multi-Spectral Imager (MSI) and a Broadband Radiometer (BBR) (Illingworth et al., 2015; Wehr et al., 2023).
The innovation of having two active (CPR, ATLID) and two passive (MSI, BBR) instruments on a single platform enables a
highly synergistic approach in characterizing the state of the atmosphere. It is an unprecedented observational setup which will
offer novel opportunities in atmospheric research beyond the initial mission goals. CPR, ATLID and MSI are used to retrieve
a three-dimensional (3D) scene (e.g., Qu et al., 2022a; Mason et al., 2022) to calculate the radiative flux which is compared
to the radiometer (BBR) measurements on board (Barker et al., 2023). The European and Canadian EarthCARE processing
chain is presented by Eisinger et al. (2023). The need to derive the radiative flux at the top of the atmosphere with an accuracy
of 10 $Wm^{-2}$ for a 100 $km^2$ snapshot view of the atmosphere is the leading idea for the EarthCARE mission requirements
(MRD, 2006). The vertical profiles of clouds and aerosol layers along the satellite track are provided by the active instruments
ATLID and CPR (e.g., van Zadelhoff et al., 2023; Donovan et al., 2023b; Kollias et al., 2022; Irbah et al., 2022). In order to get
information about the scene apart from the satellite track, the passive imager MSI is necessary which provides columnar observations over a 150 km wide swath (Docter et al., 2023; Hünerbein et al., 2022; Hünerbein et al., 2023). The idea of combining

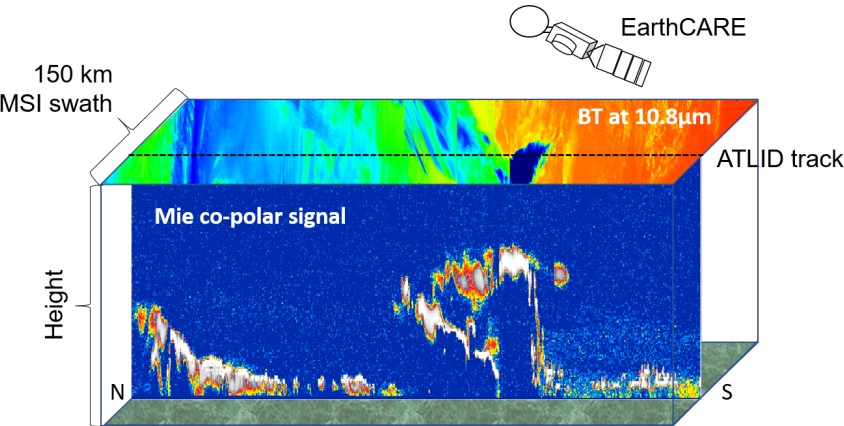

**Figure 1.** Combined view of ATLID ("curtain") and MSI ("carpet") on the simulated, so-called *Halifax* scene. A strong ATLID Mie co-polar
signal (white color) indicates optically thick clouds, weaker signals (red to yellow) indicate optically thinner clouds or aerosol layer. The high
clouds in the center of the scene are detected by MSI on the basis of their low brightness temperature (BT, blue color). The high brightness
temperatures (red color) on the MSI swath result from the surface return where the low broken clouds are visible in yellow.





the vertical information from ATLID along track ("curtain") with the columnar information from MSI along and across track ("carpet") is illustrated in Fig. 1. This combination is an important step in the synergistic approach of EarthCARE, especially in estimating the cloud top height (CTH) of optically thin clouds and in assessing the aerosol type for the entire scene. The

high-spectral-resolution lidar ATLID (do Carmo et al., 2021) operates at a wavelength of 355 nm with a vertical resolution of approximately 100 m below an altitude of 20 km and 500 m above 20 km. It provides vertical profiles along the satellite track of the particle backscatter and extinction coefficient, the lidar ratio and the particle linear depolarization ratio which are provided in the ATLID L2a product A-EBD (ATLID Extinction, Backscatter, Depolarization, Donovan et al., 2023b). The multi-spectral imager MSI measures the radiances in the visible, near-infrared and infrared (central wavelengths: 0.67, 0.865,

1.65, 2.21, 8.8, 10.8, 12.0 μm) with a 500 m spatial resolution over a swath width of 150 km across track. Combinations of these wavelengths are used to derive a cloud mask which is provided in the MSI Cloud Mask product (M-CM, Hünerbein et al., 2022) and to retrieve cloud optical properties such as the cloud optical thickness (COT), CTH and the effective radius of the cloud droplets which is provided in the MSI Cloud Optical and Physical product (M-COP, Hünerbein et al., 2023). Aerosol products such as the aerosol optical thickness are retrieved for the cloud-free pixels and stored in the MSI Aerosol Optical

Thickness product (M-AOT, Docter et al., 2023).

Regarding clouds, an accuracy of the CTH for ice and water clouds of 300 m is required (mission requirements) for the 3D scene. Such accuracy cannot be achieved with MSI retrievals alone. The MSI CTH retrieval (Hünerbein et al., 2023) is based on the measured radiation at 10.8 μm which is thermally emitted by clouds (Fritz and Winston, 1962; Smith and Platt, 1978; Wielicki and Coakley, 1981) and gives an infrared effective radiative height. The method provides a reasonable estimate for the

CTH for optically thick clouds, but in case of semi-transparent cloudiness the direct use of the measured brightness temperature will lead to a significant underestimation of the true CTH. On the other hand, ATLID can provide the physical boundaries of the cloud with the required accuracy (A-CTH product, Wandinger et al., 2023), but only for an atmospheric cross section along track. Therefore, an algorithm for a synergistic ATLID–MSI CTH product (AM-CTH) is developed and described in the present paper. The AM-CTH product is based on the systematic investigation and classification of differences in the CTH

obtained with ATLID and MSI along track. A scene classification scheme is developed to extrapolate the CTH difference to the MSI swath.

With respect to aerosol, the mission requirements demand to identify the presence of absorbing and non-absorbing aerosol particles from natural and anthropogenic sources. Vertically resolved aerosol typing is provided along track by the ATLID Target Classification (A-TC, Donovan et al., 2023b). These aerosol types weighted by the extinction coefficient of the respective

height level are integrated to a column aerosol mixture in the ATLID Aerosol Layer Descriptor (A-ALD, Wandinger et al., 2023). The dominant aerosol type can be compared to the aerosol mixing ratios applied in M-AOT. The combination of ATLID observations at 355 nm with MSI retrievals for wavelengths $\geq$ 670 nm (Docter et al., 2023) further supports the aerosol typing. The ATLID–MSI Aerosol Column Descriptor (AM-ACD) is developed as a synergistic product to combine aerosol information from the two instruments. The AM-ACD contains information on the spectral AOT, respective Ångström exponents, and an

estimate of the aerosol type.

AM-COL extends the ATLID information over the entire swath as long as a swath pixel can be related to a track pixel. A more



sophisticated approach including radiative transfer simulations is used for the pixels close to the track in the ACM-3D product (Qu et al., 2022a). They prepare the data for the 100 km$^2$ snapshot (20 km along track × 5 km across track) which will be used for the radiative closure. These simulations can be done for two pixels in each direction from the track, but not for the entire

swath. The AM-COL processor does not construct a 3D scene, but will provide the CTH and the columnar aerosol products (2D horizontally like a "carpet") for the entire MSI swath width of 150 km.

The paper is structured as follows. Sect. 2 provides an overview about previous efforts in combining active and passive remote sensing for the determination of the CTH and for aerosol typing. Then, a detailed description of the underlaying AM-COL algorithms is provided in Sect. 3. The algorithm is validated using the common test scenes from the EarthCARE End-to-End

Simulator (Donovan et al., 2023a) in Sect. 4. Cloud and aerosol products are always treated separately. Major findings are summarized in the Conclusion.

## 2 Combining active and passive remote sensing

The combination of active and passive remote-sensing techniques onboard the EarthCARE satellite is essential to reach the mission goal of deriving the radiative flux at the top of the atmosphere with an accuracy of 10 Wm$^{-2}$ for a 100 km$^2$ snapshot

view of the atmosphere. In this context, the accuracy of the CTH over the MSI swath as well as the imager-based aerosol typing needs some further discussion. This section intends to provide an overview about the current state of research of these two topics.

### 2.1 Improving passive CTH retrievals by active remote sensing

The CTH is detected from space by active and passive remote sensing. Passive retrievals use for example the MODerate-

resolution Imaging Spectrometer (MODIS) or the Spinning Enhanced Visible and InfraRed Imager (SEVIRI); and active measurements are taken with lidar from the Cloud-Aerosol Lidar and Infrared Pathfinder Satellite Observations (CALIPSO). Active remote sensing has a high vertical resolution in detecting the geometrical CTH, but is limited to observations along the narrow satellite track. Passive remote-sensing techniques offer a wider spatial coverage, but with limited vertical accuracy.

From the literature it is known that CTH retrievals from passive sensors can be highly erroneous. Comparisons with lidar

measurements showed large discrepancies in dependence on the type, height, and optical thickness of the clouds. First space-borne comparisons of CTH detection with passive and active sensors were presented by Mahesh et al. (2004) and Naud et al. (2005). These authors used lidar observations from the Geoscience Laser Altimeter System (GLAS) to assess CTH accuracy for MODIS (aboard Terra and Aqua) and SEVIRI (aboard Meteosat-8). Beside discrepancies in the cloud mask, especially over polar regions and for optically thin clouds, they observed that the passive instruments overestimate the top height of low

and opaque clouds by 0.3–0.4 km and underestimate the CTH for high and optically thin clouds. Further comparison studies (Weisz et al., 2007; Holz et al., 2008; Minnis et al., 2008; Yao et al., 2013; Iwabuchi et al., 2016) reported different biases depending on geographical region, cloud type and altitude. Major improvements to the passive retrievals were achieved by MODIS Collection six (Baum et al., 2012). ESA's Clouds Climate Change Initiative resulted in a comprehensive overview



about state-of-the-art retrievals of cloud properties from passive sensors (Stengel et al., 2015). A very detailed study with wide
spatial coverage was performed by Mitra et al. (2021). They investigated the bias of Terra-MODIS between 50°S and 50°N
against the space lidar CATS (Yorks et al., 2016) for various altitude and cloud optical thickness (COT) ranges. In the case
of high clouds (CTH > 5 km, defined by CATS), the bias (MODIS–CATS) was found to be –1.16 km (with a precision of
1.08 km), and for low clouds (< 5 km) the bias was 40 ± 730 m. Especially for low clouds, the bias strongly depends on COT:
Optically thin (COT < 0.8) low clouds showed a negative bias of –440±600 m, whereas optically thick (COT > 0.8) low clouds
were found to have a positive bias of +500±430 m (Mitra et al., 2021). For high clouds, the bias reduces with increasing COT
to –280 m for COT > 0.8. The presence of multi-layer clouds increases the bias between active and passive detection of CTH
(–1.20 ± 1.19 km).

Special care has to be taken in presence of low-level clouds in the Arctic which under certain conditions are detected with an
imager but not from the space lidar (Chan and Comiso, 2011). These clouds are frequently observed in summer (Griesche et al.,
2020) and are hardly visible by ground-based cloud radars because of their low altitude. Further challenges for passive CTH
detection occur in the presence of thick dust layers (e.g., Robbins et al., 2022). Thus, a proper aerosol–cloud discrimination is
essential.

New algorithms use machine learning or neuronal networks to obtain the CTH from passive sensors (e.g., Håkansson et al.,
2018; Min et al., 2020). These algorithms are trained on previous data sets using CALIPSO. As a recent example, Tan et al.
(2022) published an algorithm to assess the CTH of overlapping clouds from the Advanced Himawari Imager (AHI). Their
machine-learning approach uses the available information on cloud phase, COT and neighboring cloud pixels to estimate the
CTH of water and overlaying ice clouds. In a validation against CloudSAT and CALIPSO the algorithm of Tan et al. (2022)
led to a reduction of the mean CTH bias from –5.1 to –2.6 km.

## 2.2 Aerosol typing from combined active and passive remote sensing

Besides the knowledge about the aerosol optical thickness (AOT) and the aerosol layer heights, a correct aerosol typing is
essential for radiative transfer calculations. The radiative properties of an aerosol layer depend on the aerosol type or mixture.
In case of EarthCARE, the Hybrid End-To-End Aerosol Classification model (HETEAC, Wandinger et al., 2022) is the under-
lying aerosol model linking the optical, microphysical and radiative properties of aerosol mixtures.

Aerosol classification schemes from active remote-sensing observations are based on the observed (intensive) optical proper-
ties. In the case of lidar measurements, the particle linear depolarization ratio (measure of particles' non-sphericity) and the
extinction-to-backscatter ratio (lidar ratio) are the main quantities used in aerosol classification schemes (e.g., Burton et al.,
2012; Groß et al., 2015). A comprehensive data base of these intensive optical properties at 355 and 532 nm was collected
by Floutsi et al. (2022). The CALIPSO aerosol classification scheme (Omar et al., 2009; Kim et al., 2018) could not use the
lidar ratio as input because there is no direct measurement of the extinction coefficient. In contrast to CALIPSO, EarthCARE
will carry a high-spectral-resolution lidar (HSRL), which provides independent measurements of the particle extinction and
backscatter coefficients (at 355 nm) and therefore enables an improved aerosol classification. The first HSRL system operated
successfully in space was the lidar onboard of ESA's wind lidar mission Aeolus which enabled the independent measurement



of the extinction coefficient (Ansmann et al., 2007; Flament et al., 2021). In the case of multi-wavelength observations, the Ångström exponent provides additional information about the particle size. A vertically-resolved aerosol typing is only possi-
ble with active remote-sensing instrumentation.

Passive remote-sensing techniques use multiple wavelengths to retrieve the AOT. From these AOT observations and the related Ångström exponents, the columnar aerosol type is determined (e.g., Toledano et al., 2007; Holzer-Popp et al., 2013; de Leeuw et al., 2015). Including polarization measurements (e.g., Russell et al., 2014) or trace-gas column densities (Penning de Vries et al., 2015) provides additional information to improve aerosol typing. In contrast to the Ångström exponent or the polariza-
tion, the AOT is an extensive property and therefore not intrinsic to a certain aerosol type.

**Table 1.** The main input and output parameters for the ATLID–MSI Cloud Top Height product and the products (with references) in which they are contained. Dimensions: X – along track, Y – across track.

| Product name | Resolution | Dimension |
|---|---|---|
| **Input** | | |
| **ATLID L2a Cloud Top Height** (A-CTH, Wandinger et al., 2023) | | |
| – ATLID cloud top height | JSG | X |
| – Simplified uppermost cloud classification | JSG | X |
| **MSI L2a Cloud Mask** (M-CM, Hünerbein et al., 2022) | | |
| – MSI cloud mask | MSI grid | X,Y |
| – MSI cloud phase | MSI grid | X,Y |
| – Surface classification | MSI grid | X,Y |
| – M-CM quality status | MSI grid | X,Y |
| **MSI L2a Cloud Optical and Physical products** (M-COP, Hünerbein et al., 2023) | | |
| – MSI cloud top height | MSI grid | X,Y |
| – MSI cloud optical thickness | MSI grid | X,Y |
| – MSI cloud top pressure | MSI grid | X,Y |
| **MSI L1c data** | | |
| – MSI brightness temperature at 10.8 µm | MSI grid | X,Y |
| – MSI brightness temperature at 12.0 µm | MSI grid | X,Y |
| – MSI reflectance at 0.67 µm | MSI grid | X,Y |
| **Output** | | |
| **ATLID–MSI L2b Cloud Top Height** (AM-CTH, this paper) | | |
| – ATLID–MSI cloud top height difference | JSG | X,Y |
| – MSI cloud top height | JSG | X,Y |
| – Cloud fraction | JSG | X,Y |
| – AM-CTH quality status | JSG | X,Y |



## 3   ATLID–MSI Column Products processor (AM-COL)

The ATLID–MSI Column Products processor (AM-COL) produces the ATLID–MSI Cloud Top Height (AM-CTH) product and the ATLID–MSI Aerosol Column Descriptor (AM-ACD) product. These products belong to the EarthCARE L2b products defined in the ESA EarthCARE production model and product list (Wehr et al., 2023; Eisinger et al., 2023). Since their generation requires input from ATLID L2a products created in the ATLID Layer Products processor (A-LAY, Wandinger et al., 2023) and MSI L2a products created in the MSI Cloud Products processor and the MSI Aerosol Optical Thickness processor (M-CLD and M-AOT, Hünerbein et al., 2022; Hünerbein et al., 2023; Docter et al., 2023), they are produced after the complete ATLID L2a and MSI L2a processing is completed. An overview about the main input and output parameters and the respective products in which they are contained is provided for the cloud products in Table 1 and for the aerosol products in Table 2.

All calculations within the AM-COL processor are performed for one grid cell horizontal resolution on the EarthCARE Joint

**Table 2.** The main input and output parameters for the ATLID–MSI Aerosol Column Descriptor product and the products (with references) in which they are contained. Dimensions: X – along track, Y – across track, $C_4$ – MSI aerosol components, $C_7$ – ATLID aerosol types

| Parameter | Resolution | Dimension |
|---|---|---|
| **Input** | | |
| **ATLID L2a Aerosol Layer Descriptor** (A-ALD, Wandinger et al., 2023) | | |
| – Column aerosol optical thickness at 355 nm | JSG | X |
| – Columnar aerosol classification probabilities | JSG | $X,C_7$ |
| – Number of detected aerosol layers | JSG | X |
| **MSI L2a Aerosol Optical Thickness** (M-AOT, Docter et al., 2023) | | |
| – Column aerosol optical thickness at 670 nm | MSI grid | X,Y |
| – Column aerosol optical thickness at 865 nm | MSI grid | X,Y |
| – Aerosol component mixing ratios | MSI grid | $X,Y,C_4$ |
| – Homogeneity flag | MSI grid | X,Y |
| – M-AOT quality status | MSI grid | X,Y |
| **Output** | | |
| **ATLID–MSI L2b Aerosol Column Descriptor** (AM-ACD, this paper) | | |
| – Ångström exponent (355 nm /670 nm, 670 nm/865 nm) | JSG | X,Y |
| – Aerosol optical thickness at 355/670/865 nm | JSG | X,Y |
| – Dominant aerosol type | JSG | X,Y |
| – Dominant aerosol type flag | JSG | X,Y |
| – AM-ACD quality status | JSG | X,Y |

Standard Grid (JSG). The A-LAY products (A-CTH and A-ALD) are already provided on JSG with this resolution (approximately 1 km) along track (see Table 1 and 2). The MSI products (M-CM, M-COP and M-AOT) are provided on the finer resolution of the MSI grid (500 m). Thus, a re-sampling is necessary. The surrounding nine MSI pixels correspond to one JSG



pixel. A cloud fraction for each JSG pixel is calculated from the contributing MSI pixels. Only if all contributing MSI grid cells are categorized as cloud free (cloud fraction of 0%) or as cloudy (cloud fraction of 100%), the corresponding JSG pixel is set to cloud free or cloudy, respectively. The cloud mask for the MSI swath is provided in the M-CM product and it is based on threshold tests to brightness temperatures and reflectances of individual MSI channels (Hünerbein et al., 2022).

The AM-COL processor is split in the cloud processing algorithm AM-CTH (Sect. 3.1) applied to all cloudy pixels and the

aerosol processing algorithm AM-ACD (Sect. 3.2) applied to all cloud-free pixels. Aerosol layers above or below cloud layers are not considered.

## 3.1   ATLID–MSI Cloud Top Height (AM-CTH) algorithm

A flow chart for the ATLID–MSI Cloud Top Height (AM-CTH) algorithm is presented in Figure 2. It is applied to all JSG

pixels considered as cloud (cloud fraction of 100%) based on the MSI cloud mask. The main output of the AM-CTH processor is the CTH difference between ATLID and MSI. The ATLID CTH was determined using the wavelet covariance transform (WCT) method with thresholds from the ATLID Mie co-polar signal (Wandinger et al., 2023). The MSI CTH provided in the M-COP product was retrieved from an optimal-estimation-based algorithm using the visible, near-infared and thermal infared MSI measurements (Hünerbein et al., 2023).

In a first step, the synergistic ATLID–MSI CTH difference along track is calculated. The scene on the MSI swath has to be classified in order to find similar cloud conditions as along the track. The scene is classified with further input from the M-COP and M-CM products (e.g., COT and cloud phase) and from the MSI L1c data such as the reflectance at 0.67 μm and the brightness temperatures at 10.8 and 12.0 μm. Multi-layer cloud scenarios are searched in an extra step. Then, the CTH difference is transferred to the MSI swath. The similarity between a pixel on the swath to an along-track pixel is used

to assign the same CTH difference to the across-track pixel. At the end, the quality status of the product is determined (see Appendix A1).

The difference of the ATLID CTH and the MSI CTH is calculated along track (ATLID minus MSI). The CTH difference found on the track is related to the swath pixels under consideration of five criteria which are based on the previous scene classification:

1. Agreement in cloud type (ISCCP plus multi-layer class)

2. Agreement in cloud phase (water, ice, supercooled mixed-phase, overlapping cloud)

3. Agreement in surface type (water, land, desert, vegetation, snow, sea ice, sun glint)

4. Brightness temperature (10.8 μm) difference threshold

5. Reflectivity (0.67 μm) difference threshold

The cloud phase and surface type are provided in the M-CM product. The AM-CTH algorithm transfers them to JSG resolution under the condition that all contributing MSI pixels must have the same value.



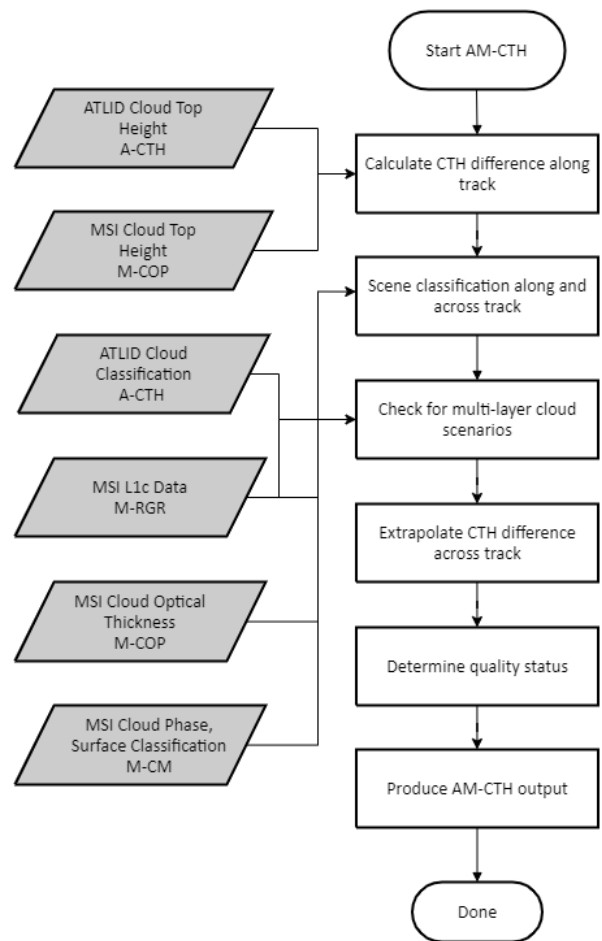

**Figure 2.** Flow chart of the ATLID–MSI Cloud Top Height (AM-CTH) algorithm. The algorithm is applied to all cloudy JSG pixels.

In order to transfer the difference detected along track to the entire MSI swath, the cloud type of each JSG pixel has be to determined. The nine cloud classes (cumulus, altocumulus, cirrus, stratocumulus, altostratus, cirrostratus, stratus, nimbostratus, deep convection) defined by the International Satellite Cloud Climatology Project (ISCCP classes, Rossow and Schiffer, 1999)

are used to categorize the cloud type of each JSG pixel. ISCCP categorizes the cloud classes by means of the cloud top pressure and the COT. From the MSI pixels contributing to one JSG pixel, the lowest cloud top pressure and the corresponding COT are used as input for classifying the JSG pixel. Both quantities are provided in the M-COP product (Table 1). Additionally, a tenth cloud class is defined as the multi-layer class. For the identification of multi-layer cloud scenarios on the MSI swath we adapt a method developed by Pavolonis and Heidinger (2004), which was used in M-CLD as well (Hünerbein et al., 2022). It makes

use of the visible reflectance ($\rho_{0.67}$) and the MSI brightness temperatures at 10.8 and 12.0 µm ($T_{10.8}$ and $T_{12.0}$). Pavolonis and Heidinger (2004) simulated brightness temperature difference ($T_{10.8} - T_{12.0}$) as function of the reflectance ($\rho_{0.67}$) in order





to set a threshold for the multi-layer-cloud detection. The combined ATLID and MSI observations along the satellite track will create an unique dataset to derive this threshold from observations. Along the ATLID track, the vertical information of ATLID easily reveals multi-layer cloud scenarios (for a semi-transparent upper cloud layer) which are flagged in the simplified

uppermost cloud classification of the A-CTH product. There, multi-layer clouds are defined when a configurable number of pixels between two detected cloud layers are cloud free (default 5 pixels, corresponding to 500 m).

Besides the agreement in cloud type, cloud phase, and surface type, two homogeneity criteria are used to determine whether the measured swath pixel can be related to a track pixel. The first criterion is based on a threshold ($\Delta T_{\mathrm{th},10.8}$) for the difference of the brightness temperature at 10.8 μm ($T_{10.8}$) between swath (s) and track pixels (t):

$$|T_{10.8,\mathrm{t}} - T_{10.8,\mathrm{s}}| < \Delta T_{\mathrm{th},10.8}. \tag{1}$$

The second criterion uses a threshold ($\Delta\rho_{\mathrm{th},0.67}$) for the difference of the MSI reflectance $\rho_{0.67}$ at 0.67 μm between swath (s) and track (t) pixels:

$$|\rho_{0.67,\mathrm{t}} - \rho_{0.67,\mathrm{s}}| < \Delta\rho_{\mathrm{th},0.67}. \tag{2}$$

The thresholds are configurable. The default values are $\Delta T_{\mathrm{th},10.8}$ = 10 K and $\Delta\rho_{\mathrm{th},0.67}$ = 0.1 based on tests with the simulated

EarthCARE test scenes (see Sect. 4).

At daytime conditions, all five criteria are used to relate a swath pixel to a track pixel. Without sunlight, there is no measurement of the reflectance at 0.67 μm, and the M-CLD algorithm cannot determine the COT and thus the cloud type. Thus, at nighttime, only three criteria (brightness temperature difference at 10.8 μm and agreement in cloud phase and surface type) are used. The quality status is set accordingly (see Appendix A1).

The search for agreement starts at the closest along-track pixel. It continues by searching one pixel before (e.g., to the North) and one pixel after (e.g., to the South) from the closest pixel along track. This alternating search is continued until an agreement is found or the configurable maximum search distance (default 75 pixels in each direction along track) is reached. If a measurement at swath fits to an along-track measurement for all criteria, then the observed CTH difference from the track is assigned to the swath pixel.

### 230 3.2 ATLID–MSI Aerosol Column Descriptor (AM-ACD) algorithm

The structure of the ATLID–MSI Aerosol Column Descriptor (AM-ACD) algorithm is illustrated in Figure 3. The algorithm is applied to all JSG pixels with a cloud fraction of 0%. The AM-ACD product contains information on the columnar aerosol optical properties. It provides the spectral aerosol optical thickness (AOT, 355 and 670 nm over land and 355, 670 and 865 nm over ocean) and the respective Ångström exponents and their uncertainties (see Table 2).

In the first step, ATLID and MSI collocated aerosol type information along track are compared (Sect. 3.2.1) and the Ångström exponent (355 nm/ 670 nm) is calculated. The ATLID AOT at 355 nm is spread over the swath in case the dominant aerosol type agrees between swath and track (Sect. 3.2.2). By investigating the horizontal homogeneity of the MSI AOT at 670 nm (identification of aerosol plumes), the ATLID aerosol typing can be spread over the entire swath or parts of it (Sect. 3.2.3).





The product contains a quality indicator which considers information on aerosol layering provided by A-ALD and an overall
quality status of the product (see Appendix A2).

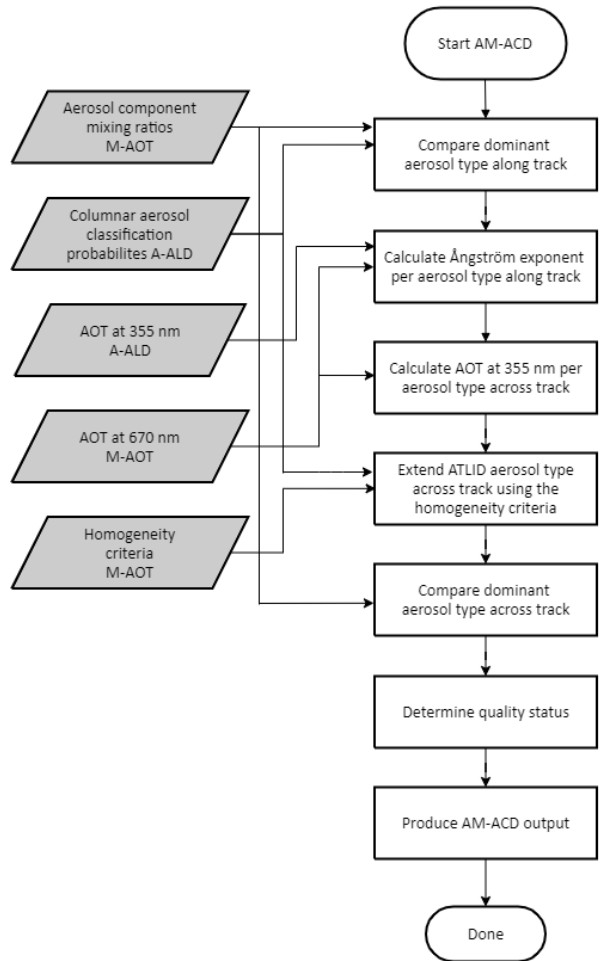

**Figure 3.** Flow chart of the ATLID–MSI Aerosol Column Descriptor (AM-ACD) algorithm. The algorithm is applied to all cloud-free JSG
pixels.

### 3.2.1   Comparison of the dominant aerosol type

In Section 2.2 the active and passive aerosol typing approaches were introduced. The ATLID aerosol typing is based on
the measurements of the linear depolarization ratio and the lidar ratio. Seven aerosol types (dust, marine aerosol, continental
pollution, smoke, dusty smoke, dusty aerosol mix, ice) are distinguished. If the aerosol type ice dominates the column integrated
aerosol classification, a cirrus cloud is included in the profile which was not detected by the A-CTH algorithm. The profile
is therefore not cloud free and a warning is raised. In the following, only the first six aerosol types (excluding the ice) are



considered for comparison. The aerosol types are provided as a vertical profile in the A-TC product (Donovan et al., 2023b) and used by the A-ALD algorithm to calculate the column-integrated aerosol classification probabilities for a better comparison with MSI. The MSI aerosol typing is based on an a priori aerosol climatology over land taken from Kinne et al. (2013) and on

a best fitting component mixture to the MSI measurements over ocean (Docter et al., 2023). The M-AOT aerosol classification uses the four aerosol components from HETEAC (Wandinger et al., 2022), which include two fine modes (weakly absorbing and strongly absorbing) and two coarse modes (spherical and non-spherical).

In Table 3, the six A-TC aerosol types are expressed in terms of the four HETEAC aerosol components. The four pure A-TC types (dust, marine aerosol, continental pollution, smoke) are clearly dominated by one the four HETEAC components. The

A-TC aerosol types dusty smoke and dusty aerosol mix are a mixture of two or three HETEAC aerosol components. Both mixtures are found for an AOT contribution of coarse-mode non-spherical (CMNS) aerosol between 25 and 50%. The more absorbing dusty smoke requires more than 20% of fine-mode strongly absorbing (FMSA) aerosol; whereas the less absorbing dusty aerosol mix should have a contribution of less than 20% of fine-mode strongly absorbing aerosol.

Along the ATLID track a direct comparison of the six A-TC aerosol types and the four HETEAC components is achieved. If A-

TC is dominated by a mixture (dusty smoke or dusty aerosol mix), the above derived thresholds are applied for the comparison with the M-AOT aerosol classification. In case of agreement, the dominant aerosol type flag is set to 1, otherwise it is 0.

**Table 3.** The representation of the six aerosol types from the ATLID target classification (A-TC, Donovan et al., 2023b) in terms of AOT contributions of the four basic aerosol components defined in HETEAC (Wandinger et al., 2022): FMWA – fine mode weakly absorbing, FMSA – fine mode strongly absorbing, CMS – coarse mode spherical and CMNS – coarse mode non-spherical. The optical properties (particle linear depolarization ratio and the lidar ratio at 355 nm) and uncertainty ranges are provided for each A-TC aerosol type.

| A-TC aerosol type | Optical properties | | AOT contribution (in %) | | | |
| | Depol. ratio | Lidar ratio (sr) | FMWA | FMSA | CMS | CMNS |
|---|---|---|---|---|---|---|
| **Dust** | 0.22±0.05 | 55±15 | 14 | 0 | 2 | 85 |
| **Marine aerosol** | 0.03±0.04 | 20±12 | 0 | 0 | 99 | 1 |
| **Cont. Pollution** | 0.03±0.04 | 55±15 | 85 | 0 | 12 | 2 |
| **Smoke** | 0.03±0.04 | 88±12 | 22 | 76 | 0 | 2 |
| **Dusty smoke** | 0.14±0.06 | 73±15 | 0 | 61 | 0 | 39 |
| **Dusty aerosol mix** | 0.14±0.06 | 43±15 | 36 | 0 | 26 | 38 |

### 3.2.2  Extrapolation of the AOT at 355 nm from the track to the swath

The idea of the AM-ACD algorithm is to extrapolate the AOT at 355 nm as measured with ATLID to the MSI swath in order to increase the aerosol information over the entire swath. Therefore, it is important to capture the spatial extent of an aerosol

plume across track and combine it with the measurements along track. ATLID observes the AOT at 355 nm, MSI at 670 and 865 nm over ocean and at 670 nm over land. The Ångström exponent describes the spectral AOT behavior. It is an aerosol-type





characteristic parameter which mainly contains information on the mean size of the particles (e.g., Toledano et al., 2007).

If the dominant aerosol type agrees (see Sect. 3.2.1), the AM-ACD algorithm calculates the Ångström exponent (355 nm/670 nm) along track. In every EarthCARE frame (1/8 orbit) the mean Ångström exponent is calculated per dominant aerosol type
(if it is present within the frame). From the MSI aerosol classification the dominant aerosol type is derived for each JSG pixel across track. In case the same dominant aerosol type was detected along track as well, the respective Ångström exponent is used to calculate the AOT at 355 nm from the MSI-measured AOT at 670 nm. An aerosol plume consisting of a dominant aerosol type which is just present on the MSI swath but not on the ATLID track cannot be handled by the AM-ACD algorithm as the information about the relationship between the two wavelengths is missing.
Alternatively, HETEAC could be used to calculate the Ångström exponent based on the aerosol component mixing ratios (from M-AOT) or the columnar aerosol classification probabilities (from A-TC, A-ALD). However, we decided to implement the described observation-driven approach in AM-ACD.

### 3.2.3   Extension of the ATLID aerosol classification to the MSI swath

The M-AOT product provides a homogeneity flag (Table 2) which indicates whether the optical properties of the surrounding
pixels are counted as homogeneous. This flag is used to transfer the dominant aerosol type derived from ATLID observations along track to the MSI swath. As long as the homogeneity criterion is fulfilled the same dominant aerosol type as derived for the closest along-track pixel could be assumed for the across-track pixel. The additional M-AOT aerosol typing provides the possibility of comparison.

A simple aerosol classification based on the AOT at 670 nm and the Ångström exponent (355 nm/670 nm) would be possible.
Passive remote-sensing techniques applied this method in the past (e.g., Toledano et al., 2007). However, we do not consider the AOT as an adequate parameter for aerosol typing because it depends on the amount of aerosol (extensive quantity) and not on the aerosol type characteristics. As an example, a thin dust layer (low AOT, low Ångström exponent) might be missclassified as marine aerosol. Here, we prefer to extend the ATLID aerosol typing to the swath. It is based on the intensive quantities of particle linear depolarization ratio and lidar ratio. To stay with the example, the higher depolarization ratio would clearly
identify the dust layer and would not lead to a confusion with marine aerosol. We leave it open to the user to construct an own aerosol classification scheme based on the columnar quantities provided (AOT at 355, 670 nm and over ocean additionally at 865 nm and the respective Ångström exponents, see Table 2).

## 4   Validation of the AM-COL processor with the EarthCARE test scenes

The synergistic AM-COL processor does only partly use L1 data from instruments but mainly combines ATLID and MSI L2a
products to generate a L2b columnar product. This fact prevents us from using real-world data for its validation. As presented in Section 2.1, MODIS-retrieved CTHs are validated against space-lidar derived CTHs. The synergistic AM-COL processor already combines active and passive remote sensing. Thus, at the present state it can be only validated against simulated test scenes available for the EarthCARE processing chain.





With the EarthCARE End-to-End Simulator specific test scenes were created to test the full chain of EarthCARE processors
(Donovan et al., 2023a). All scenes are based on the Global Environmental Multiscale (GEM) model output (Qu et al., 2022b).
The aerosol fields are taken from the Copernicus Atmosphere Monitoring Service (CAMS) model. In the following, we present
results obtained with the AM-COL processor for the so-called *Halifax* and *Halifax aerosol* scene which are shortly introduced
in the following paragraphs. A more detailed description is presented in Donovan et al. (2023a).

The *Halifax* scene presents a 5000 km long frame which starts over Greenland, crosses the eastern part of Canada and ends in
the Caribbean. The scene starts with clouds over the Greenland ice sheet followed by optically thick clouds down to 50°N. A
high ice cloud regime starting over east Canada down to 35°N is followed by a low-level cumulus cloud regime embedded in
a marine aerosol layer.

The *Halifax aerosol* scene is a short test scene representing the southern 2000 km of the *Halifax* scene. The marine aerosol has
been increased by a factor of 2.5, whereas all other aerosol types and the liquid clouds have been downscaled by a factor of
$10^{-6}$. This scene was specially designed to test the aerosol retrievals in more detail.

## 4.1 AM-CTH validation

Firstly, the output of the AM-CTH algorithm is presented (Sect. 4.1.1). Then, the output is validated against the GEM model
truth (Sect. 4.1.2) with a special discussion on cloud class and multi-layer clouds (Sect. 4.1.3).

### 4.1.1 AM-CTH output for the *Halifax* scene

The validation of the AM-CTH product is shown for the *Halifax* scene. In a first step, we compute the CTH difference (ATLID
– MSI) for all cloudy JSG pixels along the ATLID track. In Figure 4, the CTH of A-CTH and M-COP are shown together with
the AM-CTH difference for the *Halifax* scene along the ATLID track. Here, the CTH of the mixed-phase clouds north of 55°N
is seen by ATLID and MSI nearly at the same height. However, the multi-layer cloud scenario in the center (39–47°N) leads to
large differences. MSI is sensitive to the optically thick liquid-containing clouds at 5–7 km height and ATLID detects the thin
cirrus cloud at 11 km height as CTH. The optically thick cirrus cloud further south (36–39°N) is thick enough to be detected
by MSI as well.

Fig. 5 shows the MSI-derived CTH (on JSG), the synergistic ATLID–MSI CTH difference and the AM-CTH quality status.
North of 50°N, no sunlight is present (nighttime observations) leading to limitations in the M-CLD retrieval which are ac-
counted for in the quality status (Fig. 5). Here, only three out the five criteria for the track-to-swath transfer could be applied.
Cloud-free parts are shown in black for the AM-COL products. The CTH difference is color-plotted over the cloudy parts
shown in white. AM-CTH can provide a CTH for half of the cloudy JSG pixels (51%) defined by MSI. There are several
reasons: (1) The field of high cirrus clouds in the center could not be transferred for the entire swath. For the across-track
pixels > 60, no along-track pixels agreeing in all five criteria could be found within ±75 pixels in each direction to transfer the
CTH difference. (2) During the nighttime observations (> 50°N) the limited information from M-CLD and a quickly changing
cloud phase (one of the three nighttime criteria) made a transfer of the synergistic CTH difference difficult.

The large CTH differences in the center of the scene are originating from the thin cirrus above the liquid-containing clouds as



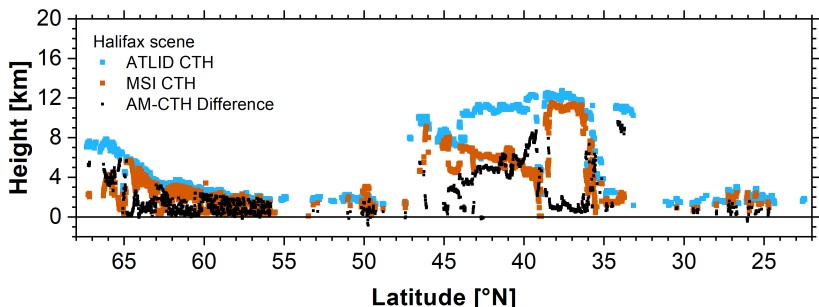

**Figure 4.** CTH along the ATLID track derived by ATLID (blue dots) and MSI (orange dots). AM-COL calculates the difference (black dots) to transfer it to the MSI swath. The results are shown for the *Halifax* scene.

seen already in the CTH difference along track (Fig. 4). The large CTH difference around 34°N is probably a misinterpretation of AM-COL due to a thin cirrus which was present along track above the low clouds. However, the mixed-phase clouds north of 55°N, the optically thick cirrus in the center and the shallow marine cumulus clouds in the South of the scene are transferred

well and the CTH difference is small (< 1 km). The algorithm performance for different cloud types is studied in more detail in Sect. 4.1.3.

### 4.1.2 CTH validation against the model truth

The results of the AM-CTH algorithm are validated against the GEM model truth (Qu et al., 2022b; Donovan et al., 2023a). In the model, the extinction coefficients for cloud water and cloud ice are provided. The central question is, how to define the

CTH from the true cloud extinction fields. Here, we will follow two distinct approaches: an extinction threshold and a cloud optical thickness (COT) threshold.

The ATLID-based approach as followed in A-CTH validation (Wandinger et al., 2023) uses an extinction threshold. The CTH is defined when the cloud extinction reaches for the first time (coming from above) a certain threshold value. In the A-CTH validation an extinction threshold of 20 $\text{Mm}^{-1}$ provided reasonable agreement between ATLID CTH and the model truth

(Wandinger et al., 2023). It provides an indication about the sensitivity of the A-CTH algorithm in detecting CTHs. This method defines the cloud as a geometrical feature and is sensitive to optically thin and thick clouds.

The MSI-based approach as followed in M-CLD validation (Hünerbein et al., 2023; Hünerbein et al., 2022) uses a COT threshold approach. Coming from above the extinction coefficient is integrated till a certain threshold COT is reached. Here, a COT threshold of 0.25 is used following the investigations of Stengel et al. (2015). They applied this threshold to CALIPSO-

derived CTHs to get a better agreement with CTHs derived from passive imagers considering the different capabilities in CTH detection. This method defines the cloud as radiative feature and is rather sensitive to optically thicker clouds.

Both methods to derive the true CTH from the GEM model truth are compared in Fig. 6. The results are shown for the 364 k (kilo – $10^3$) cloudy JSG pixels detected by the MSI cloud mask in the *Halifax* scene. The validation of the MSI cloud mask



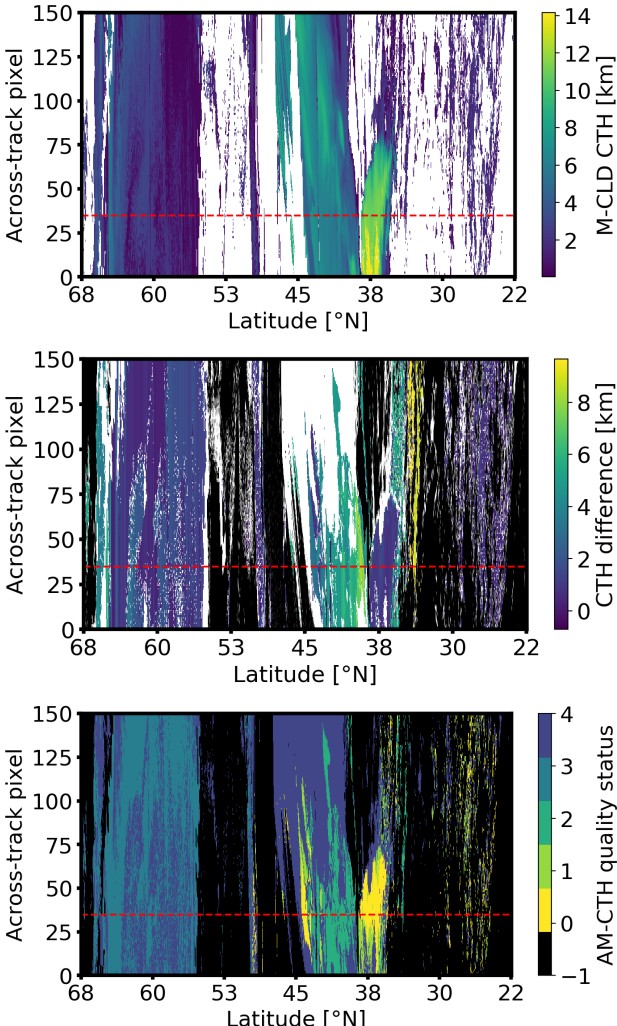

**Figure 5.** CTH for the *Halifax* scene as detected with MSI (M-CLD algorithm) on JSG (top) and the synergistic ATLID–MSI CTH difference (AM-CTH product, center). Black areas are cloud free. In the white areas M-CM detected a cloud which was not transferred by AM-CTH. The quality status of the AM-CTH product (bottom) ranging from 0 (high quality) to 4 (bad quality). A quality status of –1 is given to (cloud-free) pixels for which the AM-CTH was not applied. The ATLID track is marked with a red dashed line.

against the model truth is discussed in Hünerbein et al. (2022). In the validation of AM-COL, we are limited to the clouds
detected by M-CM on the MSI swath. From the scatter plot, it can be clearly seen that the CTH defined by an extinction
threshold of 20 Mm$^{-1}$ is always equal or higher compared to the COT threshold of 0.25. However, in 65% of the cloudy pixels
the CTH agrees within ±300 m. Especially the high clouds (>10 km height) are optically thin and reach the COT threshold of
0.25 at a lower altitude. For the validation against the model truth, we follow both CTH definitions as the best solution depends
on the research interests of the users.



The validation with the extinction threshold is shown in Fig. 7 for the MSI-alone and the ATLID–MSI retrieval as histogram

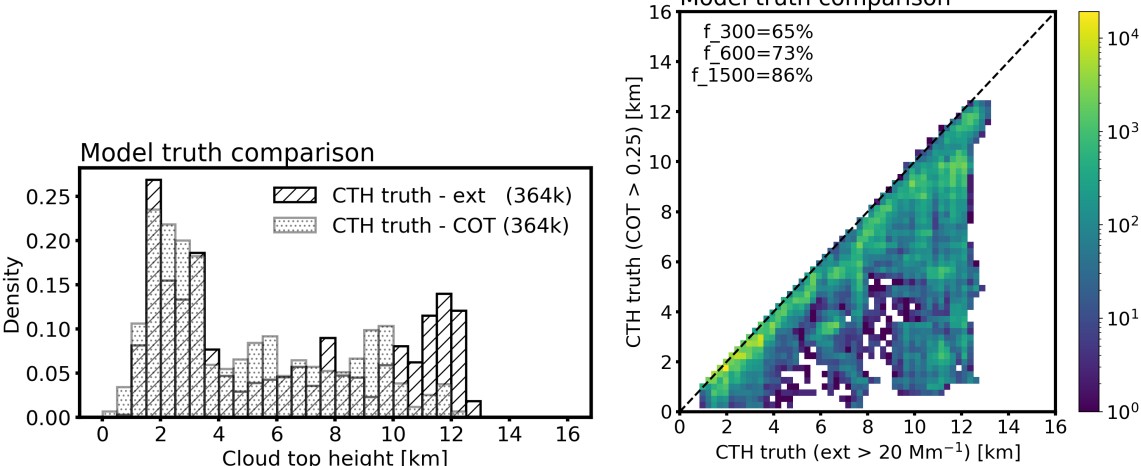

**Figure 6.** Comparison of the true CTH from GEM model output for the *Halifax* scene derived via an extinction threshold of 20 Mm$^{-1}$ (hatched) and a COT threshold of 0.25 (dotted) for all 364 kilo (factor $10^3$) JSG pixels with a cloud fraction of 100%. The indicator $f_i$ displays the percentage of data points within $\pm i$ m from the 1:1 line. The scatter plot shows that the CTH based on the extinction threshold is always higher compared to the COT threshold.


and scatter plot. M-CLD provides a CTH for 350 k JSG pixels (96%) out of the 364 k pixels detected as cloudy by the MSI cloud mask due to further quality checks in the M-COP algorithm. The AM-CTH algorithm could not assign a CTH difference for every cloud found by M-CM because several homogeneity criteria (see Sect. 3.1) have to be fulfilled to confidently translate a CTH difference from the track to the swath. Just for half of the CTHs (51%) provided in M-COP, AM-CTH can provide a

CTH. The numbers are provided with the histograms in the following figures. In case of AM-CTH, 63% of the detected CTH are within $\pm 600$ m from the 1:1 line. 40% are within $\pm 300$ m which was defined in the mission requirements. Some cirrus clouds on the swath are not detected and thus the CTH is underestimated. In some other cases, AM-CTH transferred a high (cirrus) CTH to the swath, although there were only low clouds present. Both issues occur on the swath, there just the MSI information is present. In the majority of the cases, AM-CTH captured the (geometric) CTH. The MSI stand-alone retrieval

tends to underestimate the (geometric) CTH, especially for the high clouds and some of the low clouds (see further cloud-type separated discussion in Sect. 4.1.3). Still 22% of the detected CTH are within $\pm 600$ m from the 1:1 line.

The picture changes when considering the COT-based threshold for defining the true CTH (Fig. 8). There, MSI shows a much better agreement, because the threshold is less sensitive to the thin cirrus clouds and represents the radiative CTH (Stengel et al., 2015). Now, 53% of the MSI CTHs fall within $\pm 600$ m of the 1:1 line. AM-CTH overestimates the (radiative) CTH

showing a positive bias to the 1:1 line (37% within $\pm 600$ m). Especially the cirrus clouds between 9 and 13 km height are detected by AM-CTH above a COT of 0.25.





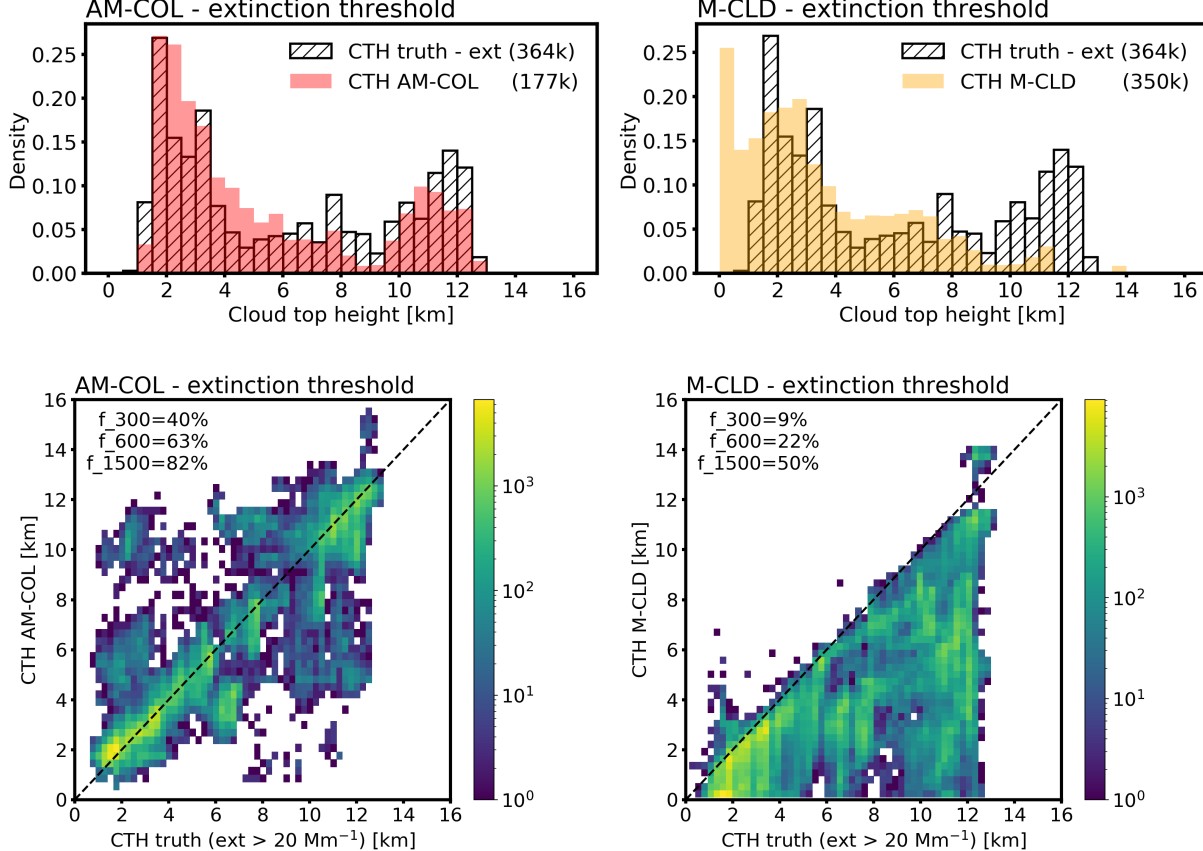

**Figure 7.** CTH validation against the GEM model truth for the *Halifax* scene. The true CTH was determined by the ATLID-based definition with a cloud extinction threshold of 20 Mm$^{-1}$ for all cloudy pixels detected by the MSI cloud mask. The histograms and scatter plots are shown for ATLID–MSI synergy product (AM-CTH, left) and MSI only product (M-CLD, right). In brackets, the total number of pixels in kilo counts for the histogram is provided. The indicator $f_i$ displays the percentage of data points within $\pm i$ m from the 1:1 line.

The amount of data points within an interval of $\pm i$ m around the 1:1 line ($f_i$ in Fig. 7 and 8) shows a similar behavior for AM-COL to extinction-based model truth (40, 63, 83% for 300, 600, 1500 m) and M-CLD to COT-based model truth (31, 53, 77% for 300, 600, 1500 m). This behavior underlines the finding that the extinction-based geometric CTH is detected by AM-COL and the COT-based radiative CTH is detected by M-CLD. In the following, we follow the extinction threshold defined CTH. A separation per ISCCP cloud type is provided in Section 4.1.3. There, a special focus is put on the multi-layer cloud scenarios.

### 4.1.3 AM-CTH algorithm performance for different cloud classes

The performance of the AM-CTH algorithm was tested for the nine ISCCP cloud classes (Rossow and Schiffer, 1999) and the multi-layer class. The detection of the latter one is mainly based on the work by Pavolonis and Heidinger (2004). Fig. 9





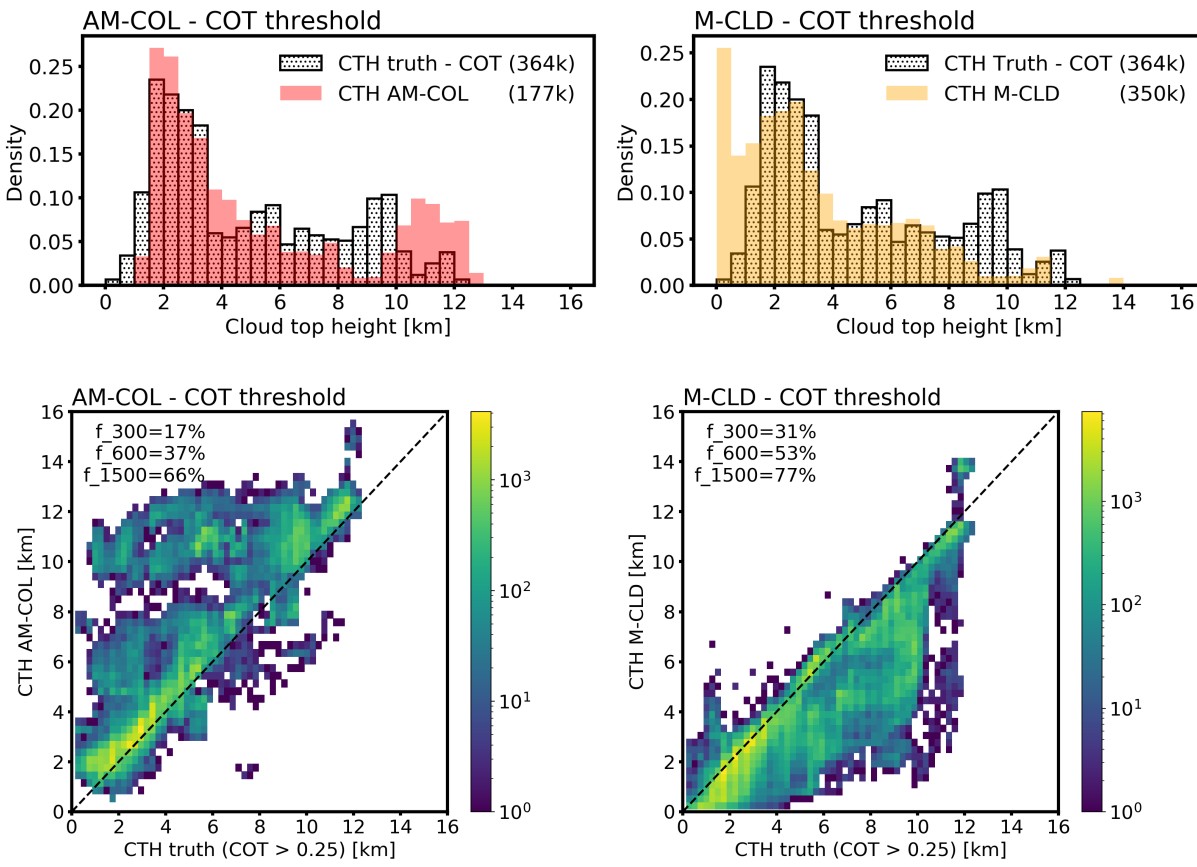

**Figure 8.** The same as Fig. 7, but here, the true CTH was determined by the MSI-based definition with COT threshold of 0.25.

presents the histograms of the CTH detected by M-CLD (orange), the synergistic CTH by AM-COL (red) and the true CTH (hatched) from the GEM model based on an extinction threshold of 20 $Mm^{-1}$ for all clouds detected by the MSI cloud mask in the *Halifax* scene. In Figure 9, the cloud class for each JSG pixel was determined by the GEM model output. The corresponding M-CLD and AM-COL results are sorted in the same cloud-class category. Best agreement between M-CLD and the model truth is reached for stratus, nimbostratus and stratocumulus clouds which are optically thick. AM-CTH is based on M-CLD and thus

agrees well with the model truth for these cloud classes. The AM-CTH algorithm improves the (geometric) CTH detection compared to M-CLD in two areas: (1) high clouds which are underestimated by M-CLD as they are too thin to be detected with MSI; and (2) cumulus and altocumulus clouds for which the CTH is detected too low by MSI. A closer inspection of the vertical profiles of the extinction in each cloud class showed that the maximum in the extinction and thus optical depth is reached much lower than the geometric CTH, especially for the optically thin clouds (left column of Fig. 9) and the high clouds

(top row of Fig. 9). In general, MSI underestimates the CTH if we consider the geometric boundaries of the cloud by applying an extinction-based threshold (Fig. 7 and 9). MSI is sensitive to the radiative boundary of the cloud (see COT-based threshold



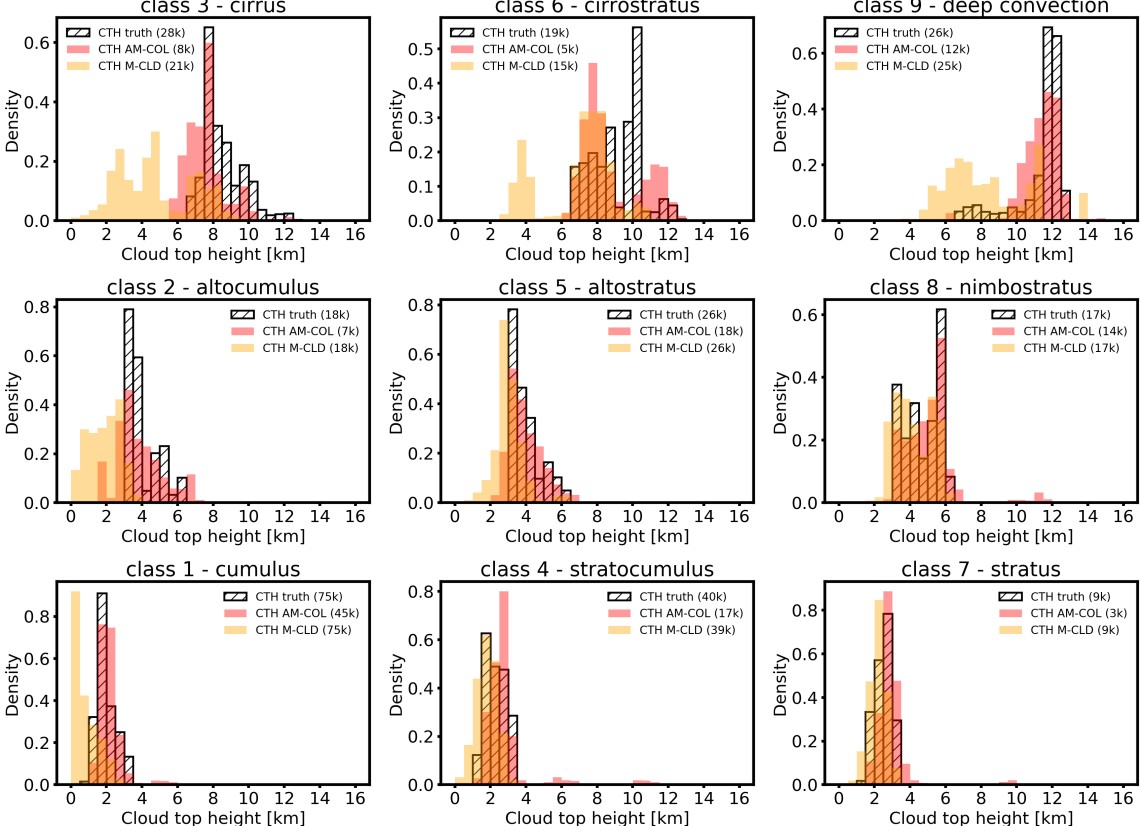

**Figure 9.** Histograms of the CTH validation against the GEM model truth (hatched) for the nine ISCCP cloud classes. The cloud class was defined by the GEM model truth using an extinction threshold of 20 Mm$^{-1}$. The multi-layer clouds are not included. The output of M-CLD (orange) and AM-COL (red) for the same pixel are presented for the *Halifax* scene. In brackets, the total number of pixels in kilo counts is provided for each cloud class.

in Fig. 8), which coincides with the geometric boundary in case of optically thick clouds such as stratus, nimbostratus and stratocumulus clouds.

The number of JSG pixels considered in the histogram is provided in the plots. As previously stated (Section 4.1.2), AM-CTH

was able to transfer a CTH difference for half of the CTHs (51%) provided in M-COP in the case of the *Halifax* scene. A special challenge are the multi-layer clouds for which the results are presented in Figure 10. The definition applied to the GEM model output follows the criteria introduced in the A-CTH algorithm (Wandinger et al., 2023) stating, that at least five height bins corresponding to approximately 500 m of clear air has to be present between two cloud layers to be classified as multi-layer. The multi-layer clouds are not included in the nine ISCCP cloud classes (Fig. 9) but treated on top as a tenth cloud class as

implemented in the AM-CTH algorithm. The multi-layer clouds are the most frequent cloud class in the *Halifax* scene with 102k JSG pixels. Figure 10 clearly shows that the CTH of the high clouds dominates the multi-layer CTH. Here, AM-COL



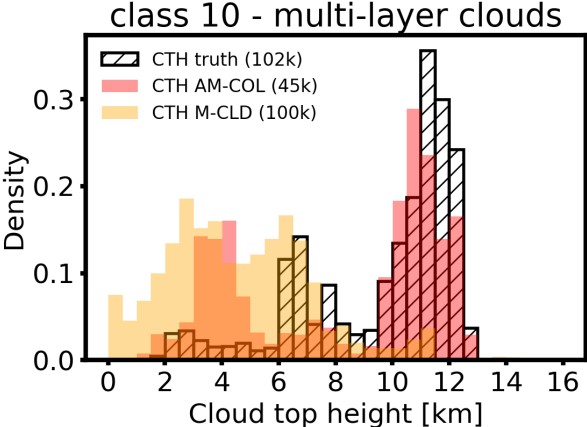

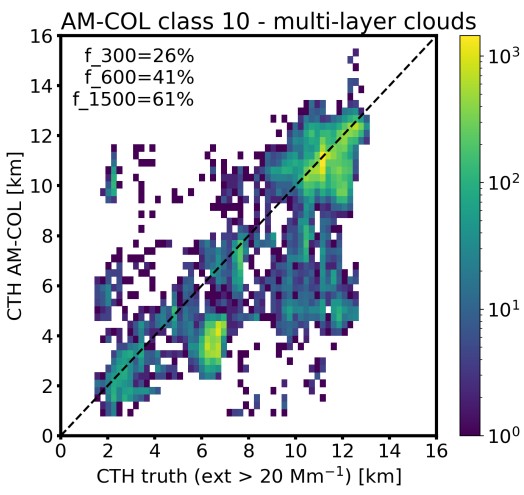
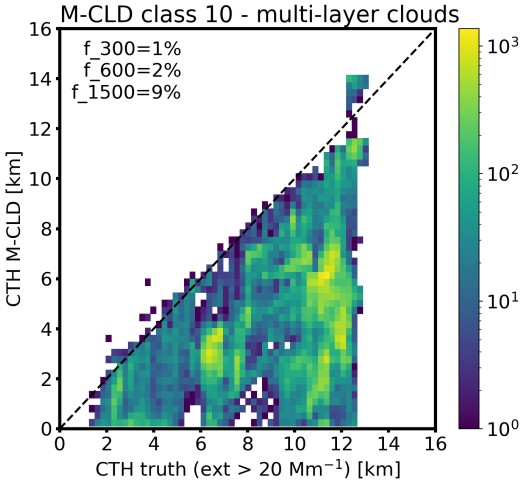

**Figure 10.** The same as Fig. 9 but for the tenth cloud class "multi-layer" (top). The scatter plots relate the CTH from AM-COL (bottom left) and M-CLD (bottom right) to the model truth based on an extinction threshold of 20 Mm$^{-1}$. The indicator $f_i$ displays the percentage of data points within $\pm i$ m from the 1:1 line.

significantly improves the (geometric) CTH detection compared to the MSI stand-along algorithm (M-CLD). 41% instead of 2% of the CTHs were detected within $\pm 600$ m from the 1:1 line. The second peak in true CTH between 6 and 8 km height is underestimated by both M-CLD and AM-COL. These clouds are further away from the track and the AM-COL CTH is based on the MSI measurements. Nevertheless, the ATLID–MSI columnar products improve the CTH detection, especially in the case of multi-layer clouds and single-layer high and optically thin clouds compared to the MSI stand-alone retrieval. MSI is sensitive to the radiative CTH rather than the geometric CTH (see Fig. 6).





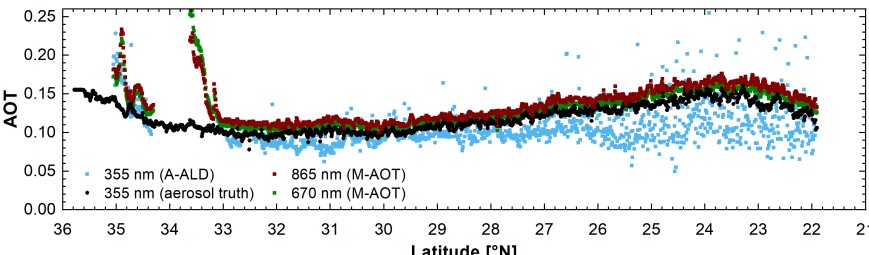

**Figure 11.** AOT along the ATLID track in the *Halifax aerosol* scene derived by ATLID (355 nm, blue) and MSI (670 nm, green and 865 nm, brown). The true AOT at 355 nm is shown in black.

## 4.2 AM-ACD validation

Firstly, the output of the AM-ACD algorithm for the *Halifax aerosol* scene is presented (Sect. 4.2.1). Then, the output is
validated against the CAMS model truth (Sect. 4.2.2)

### 4.2.1 AM-ACD output for the *Halifax aerosol* scene

The *Halifax aerosol* scene was created for the validation of aerosol retrievals and contains solely marine aerosol and some ice clouds. Thus, the dominant aerosol type for the cloud-free pixels along track was correctly classified by M-AOT and A-ALD as coarse mode spherical and marine aerosol, respectively. The AOT along track are shown in Figure 11. M-AOT provides
the AOT at 670 nm and 865 nm. A-ALD contains the AOT at 355 nm from the integrated ATLID extinction coefficient. The ice cloud at 34°N was only partly detected by the MSI cloud mask and thus the AOT of the ice crystals is included in the M-AOT product. At 35°N, the cirrus was even too thin to be detected by A-LAY, which classified the corresponding profiles as cloud free and started the aerosol retrievals (A-ALD algorithm). Here, as well the ice crystals are included in the AOT, which differs from the CAMS model truth AOT provided for aerosol only. The medium resolution output of the extinction coefficient
from the A-PRO processor (A-EBD product) was used to calculate the AOT at 355 nm. Especially in the southern part of the scene, the AOT values at 355 nm scatter a lot. The ATLID AOT in this marine-aerosol dominated scene is lower compared to the model truth by –0.0102±0.0659 for the scene <32.5°N which is not influenced by the cirrus cloud. Possible reasons for the underestimation of the AOT lie in the extinction calculation of the A-PRO processor (Donovan et al., 2023b). The high standard deviation is caused by the scattering of the ATLID AOT values. Nevertheless, the deviation from the model truth is
within the accuracy of 0.05 for the AOT as demanded by the EarthCARE mission requirements (MRD, 2006).

In the next step, the Ångström exponent (355 nm/670 nm) is calculated along track. The Ångström exponent per dominant aerosol type is obtained by averaging the Ångström exponents for all pixels along track for which the dominant aerosol type of both input algorithms (M-AOT and A-ALD) agrees. Just marine (coarse mode spherical) aerosol is present in the *Halifax aerosol* scene. An Ångström exponent for the other types is not derived as they are not present along track. The derived
Ångström exponent for marine aerosol (coarse mode spherical) is –0.23. HETEAC defines an Ångström exponent of –0.16



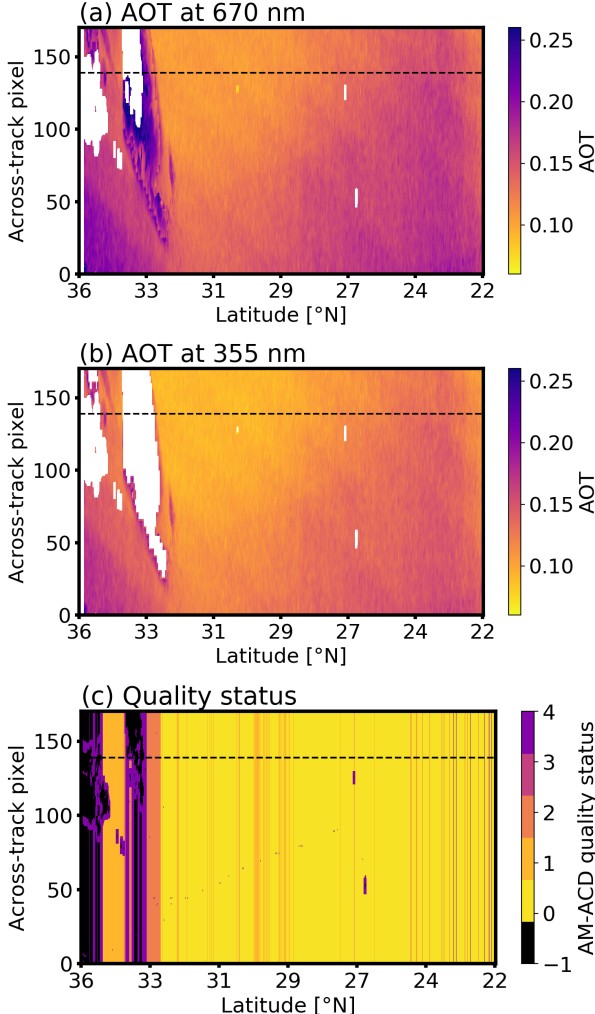

**Figure 12.** AOT at 670 nm (a) and 355 nm (b) and AM-ACD quality status (c) for the *Halifax aerosol* scene. The ATLID track is marked with a black dashed line. For the pixels categorized as cloudy, M-AOT does not derive an AOT (white areas in (a)). Still some ice cloud is present which leads to an increased AOT (>32.5°N). The M-AOT algorithm derives a different aerosol mixture for the cloud-influenced pixels. This mixture does not agree along track and therefore these pixels are not considered in the transference of the AOT at 355 nm from the track to the swath (larger white area in (b)). This behavior is reflected in the quality status of AM-ACD (details are provided in the Appendix A2).

for pure coarse mode spherical aerosol in the respective wavelength range (Wandinger et al., 2022). The too low extinction coefficient derived from ATLID and the consequently too low AOT at 355 nm is the reason for the deviation of the Ångström exponent. Nevertheless, the derived Ångström exponent is used to calculate the AOT at 355 nm on the swath from the AOT at 670 nm. The results are presented in Figure 12 together with the quality status of the AM-ACD product.




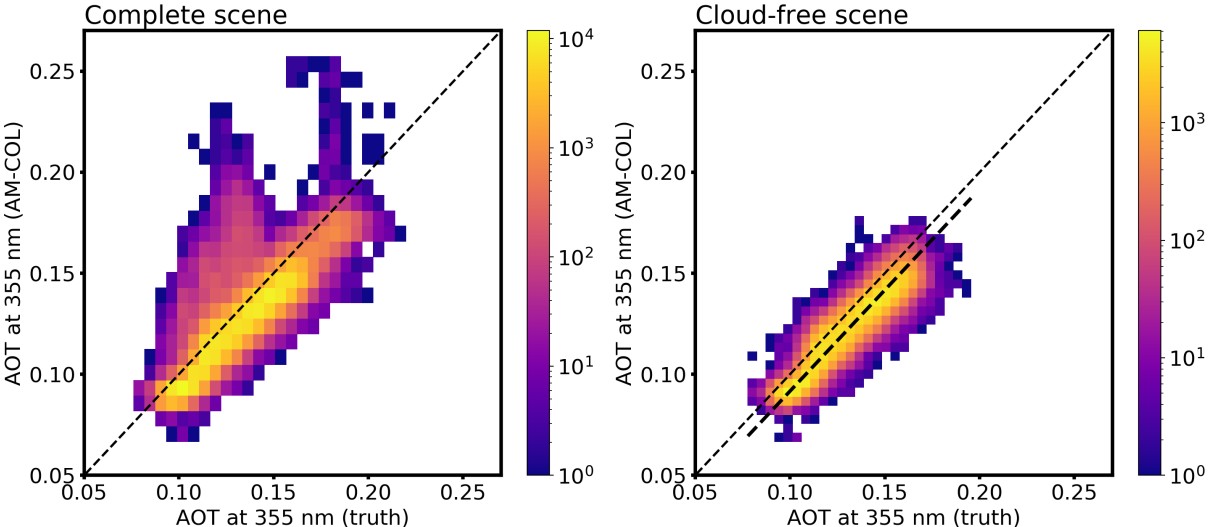

**Figure 13.** The AOT at 355 nm derived with AM-COL in the *Halifax aerosol* scene is compared against the model truth. On the left the results for the entire scene are shown. The cirrus clouds lead to an overestimation of the AOT. On the right, the scene is shown for a latitude <32.5°N, there no cloud is present (see Fig. 12). Under cloud-free conditions, the AOT is underestimated. The linear fit shown as thick dashed line indicates the mean offset of –0.0083±0.0066. The underestimation of the AOT by the ATLID products is the reason (see text for more explanations).

### 4.2.2 Aerosol product validation against the model truth

The AM-ACD products are validated against the model truth available for the simulated test scenes. The dominant aerosol type agrees for the entire scene (not shown). The AOT at 670 and 865 nm are taken from the M-AOT product (now provided on JSG) and are validated in Docter et al. (2023). The validation of the AOT at 355 nm on the MSI swath is presented in Figure 13.

The high AOT values between 0.20 and 0.25 which are not present in the model truth are caused by an incorrect aerosol-cloud discrimination. The validation is done for latitudes < 32.5°N which are not influenced by any cloud (right part in Fig. 13). The majority of the pixels follows the 1:1 line with a small negative offset of –0.0083±0.0066. The offset is caused by the negative offset of the AOT at 355 nm from upstream processors namely the extinction calculations in A-PRO (–0.0102±0.0659). In summary, the method applied in the AM-ACD algorithm itself leads to a good agreement with the model truth in the case of

the simple *Halifax aerosol* scene. Further validations of the AOT with the model truth are provided by Mason et al. (2023).

## 5 Conclusions

The synergistic ATLID–MSI Column Products (AM-COL) processor combines the strengths of ATLID in vertically-resolved profiles of aerosol and clouds with the benefits of MSI in observing the complete scene besides the track of the satellite. The uncertainties in the MSI CTH detection and MSI aerosol typing were the driving motivation to develop this synergistic L2b



algorithm. The two instruments are compared along the satellite track where they observe the same atmospheric scene. The main task of the AM-COL algorithm is to transfer this combined information from the track to the MSI swath (swath width 150 km). The algorithm is split into the analysis of cloudy pixels (AM-CTH product) and cloud-free pixels for aerosol observations (AM-ACD product) based on the MSI cloud mask.

The AM-CTH algorithm produces the synergistic CTH difference measured along the track and transfers this difference to the

swath. Several similarity criteria are used to relate an across-track pixel to an along-track pixel: agreement in cloud type, cloud phase, surface type, a brightness temperature difference (at 10.8 μm) and a reflectance difference (at 0.67 μm) threshold. For the simulated EarthCARE test scenes, it could be shown that the vertical information of ATLID improves the detection of cirrus CTHs compared to the MSI stand-alone retrieval. In addition, the CTH of cumulus and altocumulus clouds improves if ATLID input is used. The MSI retrieval underestimates the CTH of these cloud types. The usage of the simulated test scenes allows

us to study the different definitions of the CTH by using an extinction threshold or a COT threshold. The first one describes the geometric boundary of the cloud as it is seen by the lidar and the latter one describes the radiative CTH as it is seen by the imager. Special care has to be taken in case of multi-layer cloud scenarios. The improved cirrus detection of the ATLID–MSI synergy improved the multi-layer CTH determination in the simulated test scenes. However, the brightness temperature difference between 10.8 and 12.0 μm was not sensitively enough simulated to clearly detect multi-layer cloud scenarios by MSI.

Here, adaptions will become necessary once real EarthCARE data are available. The synergistic approach of a lidar and an imager on the same platform will provide insight into multi-layer cloud scenarios and their influence on passive sensors.

The AM-ACD algorithm combines the AOT observations at 355 nm from ATLID and at 670 and 865 nm from MSI to deliver an Ångström exponent. ATLID is a single-wavelength lidar and MSI has a limited amount of wavelengths at its disposal. Therefore, the Ångström exponent adds valuable input to the aerosol classification. Along track a comparison of the dominant aerosol type from MSI retrieval and the columnar aerosol classification from ATLID is possible. In case of agreement,

nant aerosol type from MSI retrieval and the columnar aerosol classification from ATLID is possible. In case of agreement, the Ångström exponent (355 nm/670 nm) is derived. It is used to transfer the AOT at 355 nm to the swath where the MSI observations at 670 nm are available. In this way, aerosol plumes are tracked from the track to the swath. The aerosol vertical distribution has an impact on the passive AOD retrieval as shown by Wu et al. (2017). EarthCARE is ideally designed to further studying this effect and to develop proper corrections based on ATLID's vertical information.

The paper describes the current stage of the AM-CTH and AM-ACD algorithms. Improvements and adaptions will become necessary once real EarthCARE data are available. Suborbital observations on the track and swath are necessary to further validate the AM-CTH and AM-ACD products during the validation phase of EarthCARE. The columnar products are designed to improve the MSI retrievals by adding the vertical and spectral information from ATLID. The combination of active and passive remote-sensing observations with close colocation will create a valuable dataset and enhance our experience for future

passive satellite missions.

*Data availability.* The simulated test data sets and the AM-COL processor outputs are available at https://doi.org/10.5281/zenodo.7117116 (van Zadelhoff et al., 2022).



*Author contributions.* UW, AH and MH designed and implemented the algorithm. MH validated it against the model truth. ND, DD and GvZ provided valuable comments on the algorithm throughout many years. SB supported the validation against the GEM model truth. MH
wrote the manuscript in strong collaboration with the coauthors.

*Competing interests.* UW is member of the editorial board of Atmospheric Measurement Techniques and co-editor of the Special Issue to which this paper contributes. The peer-review process was guided by an independent editor. The authors declare that they have no conflict of interest.

*Acknowledgements.* This work has been funded by ESA grants 4000112018/14/NL/CT (APRIL) and 4000134661/21/NL/AD (CARDI-
NAL). We thank Tobias Wehr (deceased) and Michael Eisinger for their continuous support over many years and the EarthCARE developer teams for valuable discussions in various meetings. We are grateful to Florian Schneider and Stefan Horn for the basic implementation of the code in Fortran and to Athena Floutsi for her support in describing the aerosol types in terms of HETEAC aerosol components.





## Appendix A: Quality status

### A1    Quality status of the AM-CTH product

The quality status of the cloud top height product ($Q_{CTH}$) is provided for each JSG pixel along and across track on a scale from 0 (highest quality) to 4 (bad quality). A quality status of –1 is used for JSG pixels for which no cloud was detected by M-CM. The steps of the quality status are the following:

$Q_{CTH} = 0$:    Good data, high quality. Agreement of the across-track pixel was found within $\pm 2$ pixel along track.

$Q_{CTH} = 1$:    Valid data, but agreement was found in a configurable search distance (default 75) North or South of the corresponding pixel along track.

$Q_{CTH} = 2$:    Warning: A-LAY detected multi-layer cloud scenario for the along-track pixel which was used to transfer the CTH difference to the swath.

$Q_{CTH} = 3$:    Warning: Degraded quality due to twilight or night conditions.

$Q_{CTH} = 4$:    Bad data. Observations on MSI grid are not consistent on (coarser) JSG.

$Q_{CTH} = -1$:    Not surely cloudy according to M-CM.

### A2    Quality status of the AM-ACD product

The quality status of the aerosol columnar descriptor ($Q_{ACD}$) is provided for each JSG pixel along and across track on a scale from 0 (highest quality) to 4 (bad quality). A quality status of –1 is used for JSG pixels for which a cloud was detected by M-CM. The quality status is determined along track where ATLID and MSI information is available. Using the homogeneity criteria provided by M-AOT the quality status is transferred to the MSI swath. The steps of the quality status are the following:

$Q_{ACD} = 0$:    Good data, high quality. The dominant aerosol type on track agrees between A-ALD and M-AOT. There is no significant contribution of stratospheric aerosol.

$Q_{ACD} = 1$:    Valid data, but more than a configurable number (default 2) of vertical aerosol layers were found on track.

$Q_{ACD} = 2$:    Warning: Dominant aerosol type on track disagrees between A-ALD and M-AOT.

$Q_{ACD} = 3$:    Warning: Significant amount of stratospheric aerosol is present according to A-ALD.

$Q_{ACD} = 4$:    Bad data. The homogeneity criteria of M-AOT are not fulfilled.

$Q_{ACD} = -1$:    Not surely cloud free according to M-CM.



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
