# Peer review of "Cloud top heights and aerosol columnar properties from combined EarthCARE lidar and imager observations: the AM-CTH and AM-ACD products"

_EGUsphere, 2023_

## Referee Comment (RC2)

The paper presents the AM-COL processor and AM-CTH and AM-ACD products of EarthCARE mission and evaluates their performance using simulated scenes. The paper is of high importance for the exploitation of the EarthCARE mission and falls within the scope of the AMT and the "EarthCARE Level 2 algorithms and data products" special issue. The manuscript is well structured and well written to the majority of its extent. I would suggest the publication of this work after the consideration from the authors to revise the manuscript based on the following comments/suggestions, targeted to improve the clarity of the discussions and results.

**General comment:**
Along AM-CTH and AM-ACD products, the paper presents the AM-COL processor. It would make sense to include AM-COL in the title as well.

EarthCARE Aerosol types: (a) Why in EarthCARE ice is included in the Aerosol types and not in the cloud types? (b) Why do marine and dusty mix have in their name additionally the "aerosol" wording, while not all the other aerosol types? CALIPSO has "marine" and dust mixtures types also, without the "aerosol" addition specifically for this type. Can this be harmonized for EarthCARE aerosol types also? Eg "dust, marine, continental pollution, smoke, dusty smoke, dusty mixtures"?

Because they are many processors and products discussed in the paper, it would be helpful for the reader if the abbreviations don't change during the different sections of the paper. A confusing example is the AM-CTH product which is presented in Section 3.1 and Figure 2 with this name, while later on in Section 4.1.2 it is discussed both as AM-CTH and "CTH detected by AM-COL", with its legend in the plots in fig 7 (and 8,9,10) to be "CTH AM-COL", and in Section 4.1.3 is discussed as "AM-COL CTH" or "CHT AM-COL". It is advised to describe at first from which processor each product is derived and then continue in the presentation of the flowcharts, plots, and discussions mentioning the product name (eg. AM-CTH for this case). Another case is the M-CLD or MIS CTHs in the text (eg. page 17 line 316 and 374) which in the plots is CTH M-CLD and again it would be nice to be homogenized throughout the manuscript.

**Specific comments**
**Page 1, line 20:** "Two definitions of the CTH from 20 the model-truth cloud extinction fields are compared": if there is a take-home message from this comparison, it would be interesting to be included in the abstract.

**Page 3, line 71:** "The dominant aerosol type can be compared to the aerosol mixing ratios applied in M-AOT." This is confusing, as is not clear what is done. Can this be revised to be more clear? Or else add a note for the reader that this will be presented/discussed in section 3.2.1.

**Page 3, line 71:** "The combination of ATLID observations at 355 nm with MSI retrievals for wavelengths ≥ 670 nm (Docter et al., 2023) further supports the aerosol typing." Is not very clear what/how this is used. Can you elaborate a little? Even if this will be mentioned in any of the 2 papers referred earlier, 1-2 sentence can be useful to the reader.

**Page 4, lines 94+:** Could you include the Sentinel 5P CTH retrievals in the 2.1 overview? This would be relevant to the reader who may need to use simultaneously EarthCARE and Sentinel 5P/5 for applications (e.g. for data assimilation).

**Page 5, line 142:** "wind lidar mission Aeolus": It would be nice if you could add a reference here for Aeolus mission or Aladin lidar.

**Page 7, lines 162-164:** "The A-LAY products … are already provided on JSG with this resolution (approximately 1 km) along track ... The MSI products … are provided on the finer resolution of the MSI grid (500 m)... The surrounding nine MSI pixels correspond to one JSG pixel": With 1 center pixel and 8 surrounding pixels (9 in total) of 500 m JSG would have 1.5 km resolution. How can 9 surrounding pixels of 500 m correspond to 1 km JSG along track? Maybe an explanatory diagram would clarify this question.

**Figure 2**: It would be very helpful to the reader if the flowchart is more detailed, including not only the steps followed but also the decisions in each step. So one can get from the flowchart all the information for which pixels AM-CTH data are provided and how.

**Page 10, line 227:** "(default 75 pixels in each direction along track)". Can you include here the distance in km this refers to? In MSI grid, this would mean 37.5 km along the track, in JSG grid of 1 km, this would mean 75 km.

**Page 10, lines 225-229**: The search for agreement starts at the closest along-track pixel. It continues by searching one pixel before …and one pixel after … from the closest pixel along track … This alternating search is continued until an agreement is found or the configurable maximum search distance … is reached. If a measurement at swath fits to an along-track measurement for all criteria, then the observed CTH difference from the track is assigned to the swath pixel". When reading this part is a little confusing. Only for this one swath pixel the CTH difference is assigned? And then the search for agreement stops for a more far-away grid? Please revise if it is not the case and all pixels are searched until a disagreement is found (which would be the expected case).

**Figure 3:** Same suggestion as for figure 2.

**Page 11, line 243:** "Seven aerosol types (dust, marine aerosol, continental pollution, smoke, dusty smoke, dusty aerosol mix, ice)..." This is very confusing. Why ice is in aerosol types and not in cloud types? Is this the case for the EarthCARE Aerosol type product? Why it couldn't be included in the cloud types, as is the case of CALIPSO?

**Page 11, line 243:** "Seven aerosol types (dust, marine aerosol, continental pollution, smoke, dusty smoke, dusty aerosol mix, ice)..." Why marine and dusty mix have in their name additionally the "aerosol" wording, while not all the other aerosol types? CALIPSO has "marine" and dust mixtures types also, without the "aerosol" addition specifically for this type. Can this be harmonized for EarthCARE aerosol types also? Eg "dust, marine, continental pollution, smoke, dusty smoke, dusty mixtures"?

**Page 11, line 243:** "Seven aerosol types (…) are distinguished". Here it would be useful to mention from which processor and in which product the aerosol types are provided.

**Page 11, line 244:** "If the aerosol type ice dominates the column integrated 245 aerosol classification, a cirrus cloud is included in the profile which was not detected by the A-CTH algorithm". (a) This is very confusing. If there is an ice cloud, it should be included in the A-CTH product and not be treated from the AM-CTH. And not in the Aerosol types. Why this is not the case? (b) You state that "If the aerosol type ice dominates…". If ice is present but doesent dominated, is the pixel again excluded? I believe it should be.

**Page 12, of section 3.2.1 and Table 3:** With the description provided on this page for section 3.2.1, it is not clear how the comparison will reach agreement or not. Can Table 3 be enhanced with the used thresholds of the agreement for each A-TC type? Also, can one column with the M-AOT aerosol classification be included in the Table?

**Page 13, line 268:** "If the dominant aerosol type agrees (see Sect. 3.2.1)". It would be helpful in this section to mention how the dominant aerosol type is defined. Eg., the M-AOT HETEAC component with the biggest %?

**Figure 6**: How y axis density is calculated? Scaled to the total number of pixels for every case, with 1 as a cumulative sum? Maybe is worth mentioning it. Also, the colorbar in model truth comparison plot (and relevant plots from there on) can use a legend/units (eg. nr pixels).

**Page 17, lines 375-376**: "Especially the cirrus clouds between 9 and 13 km height are detected by AM-CTH above a COT of 0.25". This is confusing. From Figure 6 I would conclude that the cirrus clouds between 9 and 13 km height are detected by AM-CTH below a COT of 0.25. But maybe there is something else you wanted to highlight. Please rephrase to make it clear.

**Page 18, line 378**: "The amount of data points within an interval of ±i m around the 1:1 line (fi in Fig. 7 and 8) shows a similar behavior for AM-COL to extinction-based model truth (40, 63, 83% for 300, 600, 1500 m) and M-CLD to COT-based model truth (31, 53, 77% for 300, 600, 1500 m)". Earlier in the manuscript (page 17 line 366) was mentioned that "40% are within ±300 m which was defined in the mission requirements". Does the statistics on page 18 show us that only the AM-COL is within the mission requirements, while the M-CLD isn't? Please consider if you would like to highlight it in this part of the paper.

**Page 22, line 418:** "Thus, the dominant". Why thus? Could it be the case that the classifications are not so successful, hence "thus" is not correct? Or there is a connection between the simplicity of the scene and the fact that the classifications are successful? If possible, modify the text to make it clear.

**Page 22, lines 420-422:** "The ice cloud at 34◦N was only partly detected by the MSI cloud mask and thus the AOT of the ice crystals is included in the M-AOT product". One wouldn't expect to find ice OD in AOT products. Why this is not the case for EarthCARE products?

**Page 22, line 420-422:** "Here, as well the ice crystals are included in the AOT, which differs from the CAMS model truth AOT provided for aerosol only". Please revise to improve the syntaxis.

**Figure 12:** Can you comment on why some values (with the highest AOT) are flagged out in the 355 nm AOT, although some seem to have quality status = 0?

**Page 22-23, lines 434-439:** "The derived …at 607 nm". Is there an error estimation for this new product (AM-ACD AOT 355)? If yes, does it consider/include the uncertainty due to the Ångström exponent bias mentioned?

**Page 35, lines 469-470:** "However, the brightness temperature difference between 10.8 and 12.0 μm was not sensitively enough simulated to clearly detect multi-layer cloud scenarios by MSI." I believe that the brightness temperature sensitivity is not discussed earlier when the results from the multi-layer cloud scenarios are presented. It would be interesting to include a comment on this in the earlier session also.

Page 27: "$Q_{CTH}$ = 4: Bad data. Observations on MSI grid are not consistent on (coarser) JSG". Coarser JSG is not defined in the text. Can you define it here?

**Technical corrections/suggestions (bold text & red ",:"):**
Page 1, line 1: "is **a** combination of multiple active…".
Page 1, line 6: "characterize the 3-dimensional scene", a suggestion to change to "characterize **a** 3-dimensional scene", or "characterize the 3-dimensional scene**s**".
Page 1, line 7: "(A-LAY)**,**   MSI L2a data from the MSI Cloud Products processor (M-CLD)**,**  the MSI Aerosol Optical Thickness processor (M-AOT)**,** as well as MSI Level 1c (L1c) data are used as input to produce the synergistic columnar products".
Page 1, line 14: "CTH detected by ATLID **and retrieved/provided** MSI is calculated": retrieved or provided by MSI is a more representative term for this product.
Page 1, line 18: "The quality status depending on day/night conditions or the presence of multiple cloud or aerosol layers is provided with the products". The syntax could be improved.
Page 2, line 34: "**a** three-dimensional (3D) scene**s** (e.g., Qu et al., 2022a; Mason et al., 2022) to calculate  radiative flux**es** which  **are** compared..".
Page 2, line 40: "about the scene  **arround** the satellite track".
Page 3, line 46-48: "It provides vertical profiles along the satellite track of the particle backscatter and extinction coefficient, the lidar ratio**,** and the particle linear depolarization ratio which are  **stored** in the ATLID L2a product A-EBD". Suggestion because they are 2 provided in 1 sentence and is less clear.
Page 3, line 52-53: "et al., 2022)**,** and to retrieve cloud optical properties such as the cloud optical thickness (COT), CTH and the effective radius of the cloud droplets which  **are** provided in the MSI Cloud Optical and Physical product".
Page 3, line 56: "for  **a** 3D scene". Is it a requirement for this scene presented in the paper, or for overall the scenes? If the 2nd "a" is needed there.
Page 3, line 9:"**a** reasonable estimate**s**".
Page 4, line 84: " using  common test scenes".
Page 4, line 86: "Conclusion**s**"
Page 4, line 96: "with lidar**s as for example** from the Cloud-Aerosol Lidar and Infrared Pathfinder Satellite Observations": CALIPSO is not the only lidar that has been used for CTH detection.
Page 4, line 100: "in dependence  **of** the type".
Page 4, line 105: "the CTH  **of** high".
Page 5, line 119: "not from  **a** space lidar".

Table 1: "and the products  in which they are contained **(bold, with references)**".

Page 7, line 158: "…after the  ATLID L2a and MSI L2a processing is completed".

Page 8, line 180: " **Then** the scene…".

Page 10, line 234: "over ocean)**,**  the respective Ångström exponents**,** and their uncertainties".

Page 14, lines 299-300: "**And more specifically w**ith the EarthCARE End-to-End Simulator specific test scenes **which** were created to test the full chain of EarthCARE processors". Something is missing in this sentence. A possible suggestion.

Page 14, line 327: "There are several reasons:": It would read better if you clarify after this text the reasons. Eg. "reasons of failure" "reasons AM-CTH can't be retrieved".

Page 15, line 340: "The central question is**:** how to define the CTH from the true cloud extinction fields**?**".

---

## Author Comment (AC1)

**Response to the review by Anonymous Referee #1**

Firstly, we want to thank you for your time and effort put into the review of our manuscript. Your careful reading and expert comments are highly appreciated and definitely improve the manuscript.

The main changes in the manuscript are the following:

- The new Fig. 2 presents a sketch to map the MSI grid to JSG and another one which illustrates the search across track by AM-CTH.
- Improved flow charts (Fig. 3 and 4)
- A new subsection describing the AM-ACD validation with *Hawaii* scene. The scene is more complex compared to the rather simple *Halifax aerosol* scene.
- The definition of the quality status of the AM-ACD product was revised.

The answers to your comments are given in bold. In the attached revised version of the manuscript our changes are marked in bold to make it easier for you to spot the new text. For the official revised version, we refer to the manuscript which was separately uploaded to AMT.

**General comments:**

This paper introduces the interesting method to transfer the observed cloud and aerosol parameters at the collocated observation of lidar and imager to the imager-only observation pixels developed by using the model simulation results for the EarthCARE satellite mission. I recommend the publication of this manuscript after the revision of the following two points.

1. The transfer method of cloud top height difference uses five criteria, but the influences of the five criteria on the results are not clear. The quality status in Figure 5 is not enough clear to understand it. Please show the figures of cloud type, cloud phase, surface type, brightness temperature, and reflectance.

We agree that more input is needed to better understand the scene. We followed your suggestion and included a new Figure 6 showing the five input quantities used to transfer the CTH difference to the swath. An explaining text was added as well:

Lines 334-341: Figure 6 presents the five quantities needed to transfer the CTH difference from the track to the swath. The reflectivity (Fig. 6e) cannot be measured at night time and the cloud type (Fig. 6a) is not retrieved for night-time or twilight conditions (>50°N). Then, only the remaining 3 criteria can be applied. During night-time, the cloud phase retrieval (Fig. 6b) alternates between ice and supercooled mixed phase clouds. Only, if all contributing MSI pixel show the same cloud phase, a cloud phase value is assigned to the JSG pixel. Otherwise no CTH difference is not transferred for the JSG pixel. It results in white spots in Figure 7b and decreased quality status. The bright ness temperature at 10.8  $\mu$ m (Fig. 6d) provides information about the scene at day and night and is therefore a valuable input parameter. The surface (Fig. 6c) does not depend on the cloud properties. The criterion of the same surface is rather conservative to be sure that only similar MSI pixels are used for the track-to-swath method.

2. The transfer method of aerosol parameters is only demonstrated in the marine aerosol case. This is an ideal situation. Please show the result when several types of aerosols exist or discuss about the scope of application of this method.

You are right, the Halifax aerosol scene represents an ideal situation. It was hard to find good aerosol conditions in the simulated test scenes because the main purpose

of the test scenes were the cloud retrievals. However, we have chosen the southern part of the Hawaii scene to show it in the paper. Although thin cirrus layers hamper a good aerosol retrieval, it is still the best that we can show at the moment. A new subsection 4.2.2 and two new figures Fig. 15 and 17 were added.

**4.2.2 AM-ACD output for the Hawaii scene**

More aerosol types are present in the Hawaii scene which will be shown to demonstrate the performance under complex aerosol situations. The dominant aerosol type shown in Figure 15a was derived from the M-AOT aerosol mixing ratios as described in Section 3.2.1. Most of the scene is dominated by fine mode aerosol which is classified as smoke, continental pollution and dusty smoke because of similar optical properties. Only south of 16°S, marine aerosol dominates. A wide area on the northern Hemisphere is affected by sun glint which leads to an increased uncertainty in the M-AOT product. In these areas, the quality status of AM-ACD is 3 as seen in Figure 15c. Thus, in the following we focus on the Southern hemispheric part of the Hawaii scene. The obtained AOT at 355 nm is presented in Figure 15b. The comparison with the model truth is provided in the next subsection.

**And in the next subsection (lines 489-497):**

In the case of the Hawaii scene, the agreement is less good. In Figure 17, we compare first the AOT at 670 nm against the model truth and see that the majority of the pixels follow the 1:1 line. The comparison is restricted to the southern hemisphere and an AM-ACD quality status of 0. The overestimation of the AOT at 670 nm by the M-AOT algorithm is caused by thin cirrus clouds which are not detected by M-CM. Therefore, these pixels are processed by the aerosol algorithm and lead to an increased AOT. AM-ACD uses the AOT at 670 nm to calculate the AOT at 355 nm on the swath. Therefore, this overestimation continues in the AM-ACD product. Moreover, the overestimation increases for the AOT at 355 nm. A mean offset of 0.054 (indicated by the dashed line) was found under these complex aerosol conditions. It is slightly above the mission requirements of 0.05. The main reason for the overestimation can be attributed to the presence of thin cirrus clouds which could not be detected by the MSI cloud mask.

**Specific comments:**

Figure 2: It is not clear to me to understand which product is input to the processing of "scene classification" and "check for multi-layer cloud". The solid arrows in figure are overlapped.

**Figure 2 (now Fig. 3) was improved to clearly show which decisions are made by the algorithm. Overlapping lines are eliminated.**

Line 196: Does "same value" mean same cloud phase and surface type?

Thank you. The statement was corrected to avoid misunderstandings. It is now clearly stated:

Lines 200-201: The AM-CTH algorithm transfers them to JSG resolution under the condition that all contributing MSI pixels must have the same cloud phase or surface type, respectively."

Line 244: Is ice included in the aerosol classification? However, ice means cirrus in the sentence. Please add an appropriate explanation.

The aerosol type definition (including ice) is described in Section 2.3.3 of the AC-TC paper (Irbah et al., 2023). The ice is considered to indicate the presence of optically thin ice-containing layers (e.g., diamond dust, subvisible cirrus) that have not been identified as clouds and thus occur in the aerosol products (statement from the A-LAY paper, Wandinger et al., 2023b). Following your question which might be the case for many readers, we add a better description:

Lines 249-257: Six aerosol types (dust, marine aerosol, continental pollution, smoke, dusty smoke, dusty aerosol mix) and ice are distinguished in the A-TC product (Irbah et al., 2023). The ice is considered to indicate the presence of optically thin ice-containing layers (e.g., diamond dust, subvisible cirrus) that have not been identified as clouds and thus occur in the aerosol products (Irbah et al., 2023; Wandinger et al., 2023b). If the aerosol type ice amounts to a significant contribution (> 20% in terms of AOT, configurable) of the column integrated aerosol classification, a cirrus cloud is included in the profile which was not detected by the A-CTH algorithm. The profile is therefore not cloud free and a warning is raised (see quality status in Appendix A2). In the following, only the six aerosol types (excluding the ice) are considered for comparison between ATLID and MSI aerosol classifications

Lines 304-310: It is difficult to understand the aerosol and cloud distributions. Please add the figures of simulation true results of Halifax scenes or refer to the figure number in Donovan et al. 2023a.

You are right, this paragraph is hard to understand without the figures of the true Halifax scene. We decided to shorten the description at this point and refer specific figures in Donovan et al., 2023a and Wandinger et al., 2023b.

Lines 317-319: A detailed description is presented in Donovan et al. (2023b), especially in Sections 3.1, 3.3 and 3.4. Furthermore, we want to refer to the plots of the ATLID Mie co-polar signal and the CTH in Wandinger et al. (2023b), there the Halifax scene is shown in Figure 6 and the Halifax aerosol scene in Figure 9.

Line 317: Does this sentence only focus on the cloud at 2-3 km altitude at 55°N? Half of CTH differences are lower than 1 km, while the other half are about 2 km between 55°N and 60°N. A more detailed discussion is needed.

You are right that this issue was not well discussed. Night-time conditions north of 50°N limit the MSI abilities and lead to less clear results from the imager. The paragraph was completely rephrased:

Lines 327-333: The CTH difference is small for the scattered clouds in the South (<  $32^{\circ}N$ ) and for the optically thick cirrus cloud at  $36-39^{\circ}N$ . However, the multi-layer cloud scenario in the center ( $39-47^{\circ}N$ ) leads to large differences. MSI is sensitive to the optically thick liquid-containing clouds at 5–7 km height and ATLID detects the thin cirrus cloud at 11 km height as CTH. Further north (>50°N), night-time conditions limit the abilities of MSI to detect the CTH. Nevertheless, the agreement is mostly within 2 km, except for the high clouds north of 65°N.

Figure 5: This quality status means the summarized information of five criteria. The influence of the five criteria is not clear. In addition, the differences of results between day and night are not discussed. See also general comment 1.

In the revised version, we clearly address the influence of the 5 criteria and we strengthened the point with the night-time observations. See our answer to your general comment 1.

Line 387: Is the cloud class in Figure 9 determined by using an extinction threshold of 20Mm-1, pressure, and temperature?

The cloud class in Figure 9 (now Figure 11) was determined by the CTH and COT of the GEM model output. Here, the CTH determined by an extinction threshold (20 Mm-1). To be consistent, only the pixels detected as cloudy by M-CM were used for these histograms. If we would take all clouds determined by the extinction threshold, much more clouds would appear in the truth. Here, we do not want to evaluate the MSI cloud mask, but the CTH.

Lines 412-414: In Figure 11, the cloud class for each JSG pixel was determined by the GEM model output (CTH determined with an extinction threshold of 20 Mm-1 and COT) only for the pixels detected as cloudy by M-CM.

Figure 11: Is the true AOT shown regardless of clouds? How about true AOT at 670 and 875 nm? Is the wavelength dependency of marine AOT small?

The true AOT is taken from the model truth and therefore it is by definition without clouds. The AOT derived from M-AOT is derived for all pixel classified as cloud-free by M-CM. Especially thin cirrus clouds are not found by the MSI Cloud Mask and therefore these pixels are included in the aerosol processing as it is the case in the northern part of the Halifax aerosol scene. The wavelength dependence of marine aerosol is based on HETEAC (Wandinger et al., 2023a). In the assets of the HETEAC paper (https://doi.org/10.5281/zenodo.7732338) a table with the aerosol optical properties at different wavelengths is provided.

The caption of Fig. 11 (now Fig. 13) was amended:

The true AOT at 355 nm is shown in black for the aerosol only regardless of the clouds above.

Line 422: The overestimation of AOT is about 0.05. It is possible to detect thin cirrus. Please show the figure of attenuated backscatter and discuss this issue.

A-CTH detects most of the thin cirrus, but mises some small parts in the north. The configurable thresholds in A-CTH might be adapted to detect more thin cirrus clouds. However, then some of the thick lofted aerosol layers might be misclassified as clouds. The current threshold settings are the optimum for the simulated test scenes and have to be adapted once real EarthCARE data are available.

For MSI it is not possible to detect the very thin cirrus clouds.

The figure of the Mie co-polar of the Halifax aerosol scene is already shown in the A-LAY paper (Wandinger et al., 2023b, Fig. 9). As the present paper and the A-LAY paper are strongly linked to each other, we do not repeat the figure but referred to it.

In the caption of Fig. 11 (now Fig. 13) we added the statement:

The ATLID scene, i.e., the Mie co-polar signal is shown in Fig. 9 of Wandinger et al. (2023b).

Figure 12: Why are the patterns of AOT and quality status different? Is AOT at 355 nm estimated in the region which the values of quality status are -1 and 4?

I agree that the quality status of the AM-ACD product is not well defined. We changed the determination of the quality status in the algorithm. Now, the quality status in Figure 13c reflects better the output of the algorithm.

The revised formulation of the quality status can be found in the appendix:

- QACD = 0: Good data, high quality of M-AOT input.
- QACD = 1: Warning: A significant amount of ice (> 20% (configurable) in terms of AOT) was detected by A-TC (provided in A-ALD). This warning is provided along track only, but probably holds for the close swath pixel as well.
- QACD = 2: Warning: Dominant aerosol type on swath was not present along the track, AOT at 355 nm could not be calculated.
- QACD = 3: Warning: The homogeneity criteria of M-AOT are not fulfilled.
- QACD = 4: Bad data. Observations on MSI grid are not consistent on JSG.
- QACD = -1: Not surely cloud free according to M-CM.

Line 449: Do you plan to examine the other cases. See general comment 2.

We examined all 3 simulated test scenes with the AM-COL processor. However, for aerosol retrievals, the dedicated Halifax aerosol scene is the best option. In the other scenes the aerosol load quite low or over land (where M-AOT uses climatological values). Nevertheless, we added the Hawaii scene for the AM-ACD validation (see answer to your general comment 2).

Line 450: Which AOT product is validated?

**We clarified the statement and have written:**

Lines 501-502: The AOT validation at 355 or 670 nm across all simulated test scenes for various processors (e.g., A-EBD, M-AOT and ACM-CAP) is provided in chapter 3.4 of Mason et al. (2023a).

**Minor comments:**

Line 38: clouds and aerosol layers --> cloud and aerosol layers

**Corrected.**

Line 158: after the complete ATLID L2a ... --> after the ATLID L2a ...

**Thank you. Corrected.**

Line 193-194: The phrases of criteria 4 and 5 are different from those of criteria 1, 2, and 3. Please rephrase them. For example, the phrase of criterion 4 is written as "Satisfaction of the criterion of the brightness temperature (10.8 mm) difference threshold (Equation 1)."

**Thank you for the helpful comment. We followed your suggestion.**

Line 435: Ångström exponent usually has positive value. Does Ångström exponent in the sentence include minus sign of a negative power?

The Ångström exponent depends on the wavelength dependence. Negative Ångström exponents are possible in the case that the AOT is higher at higher wavelengths. In the case of mineral dust, negative extinction Ångström exponents have been observed, e.g., in Veselovskii et al., ACP 2016 (Fig. 7). Furthermore, the HETEAC calculations (Wandinger et al., 2023a and assets <a href="https://doi.org/10.5281/zenodo.7732338">https://doi.org/10.5281/zenodo.7732338</a>) include the negative Ångström exponents for large spherical aerosol particles.

Veselovskii, I., Goloub, P., Podvin, T., Bovchaliuk, V., Derimian, Y., Augustin, P., Fourmentin, M., Tanre, D., Korenskiy, M., Whiteman, D. N., Diallo, A., Ndiaye, T., Kolgotin, A., and Dubovik, O.: Retrieval of optical and physical properties of African dust from multiwavelength Raman lidar measurements during the SHADOW campaign in Senegal, Atmos. Chem. Phys., 16, 7013–7028, https://doi.org/10.5194/acp-16-7013-2016, 2016.

**Cloud top heights and aerosol columnar properties from combined EarthCARE lidar and imager observations: the AM-CTH and AM-ACD products**

Moritz Haarig1, Anja Hünerbein1, Ulla Wandinger1, Nicole Docter2, Sebastian Bley1, David Donovan3, and Gerd-Jan van Zadelhoff3

[revised manuscript text omitted]

---

## Author Comment (AC2)

**Response to the review by Eleni Marinou**

**Firstly, we want to thank you for your time and effort put into the review of our manuscript. Your careful reading and expert comments are highly appreciated and definitely improve the manuscript.**

**The main changes in the manuscript are the following:**
- **The new Fig. 2 presents a sketch to map the MSI grid to JSG and another one which illustrates the search across track by AM-CTH.**
- **Improved flow charts (Fig. 3 and 4)**
- **A new subsection describing the AM-ACD validation with *Hawaii* scene. The scene is more complex compared to the rather simple *Halifax aerosol* scene.**
- **The definition of the quality status of the AM-ACD product was revised.**

**The answers to your comments are given in bold. In the attached revised version of the manuscript our changes (except some minor grammatical changes) are marked in bold to make it easier for you to spot the new text. For the official revised version, we refer to the manuscript which was separately uploaded to AMT.**

*The paper presents the AM-COL processor and AM-CTH and AM-ACD products of EarthCARE mission and evaluates their performance using simulated scenes. The paper is of high importance for the exploitation of the EarthCARE mission and falls within the scope of the AMT and the "EarthCARE Level 2 algorithms and data products" special issue. The manuscript is well structured and well written to the majority of its extent. I would suggest the publication of this work after the consideration from the authors to revise the manuscript based on the following comments/suggestions, targeted to improve the clarity of the discussions and results.*

*General comment:*
*Along AM-CTH and AM-ACD products, the paper presents the AM-COL processor. It would make sense to include AM-COL in the title as well.*

**There is the general agreement throughout the EarthCARE special issue to include the product names rather than the processor names in the title. The processors are important for the processing chain and the algorithm developers. However, the main audience of the paper is expected to work with the products and seeks information about how the product was calculated.**

*EarthCARE Aerosol types: (a) Why in EarthCARE ice is included in the Aerosol types and not in the cloud types? (b) Why do marine and dusty mix have in their name additionally the "aerosol" wording, while not all the other aerosol types? CALIPSO has "marine" and dust mixtures types also, without the "aerosol" addition specifically for this type. Can this be harmonized for EarthCARE aerosol types also? Eg "dust, marine, continental pollution, smoke, dusty smoke, dusty mixtures"?*

**(a) The aerosol type definition (including ice) is described in Section 2.3.3 and 2.3.4 of the AC-TC paper (Irbah et al., 2023). The ice is considered to indicate the presence of optically thin ice-containing layers (e.g., diamond dust, subvisible cirrus) that have not been identified as clouds and thus occur in the aerosol products (statement from the A-LAY paper, Wandinger et al., 2023b). Following your question which might be the case for many readers, we add a better description:**

**Lines 249-257: Six aerosol types (dust, marine aerosol, continental pollution, smoke, dusty smoke, dusty aerosol mix) and ice are distinguished in the A-TC product (Irbah et al., 2023). The ice is considered to indicate the presence of optically thin ice-containing layers (e.g., diamond dust, subvisible cirrus) that have not been identified as clouds and thus occur in the aerosol products (Irbah et al., 2023; Wandinger et al., 2023b). If the aerosol type ice amounts to a significant contribution (> 20% in terms of AOT, configurable) of the column integrated aerosol classification, a cirrus cloud is included in the profile which was not detected by the A-CTH algorithm. The profile is therefore not cloud free and a warning is raised (see quality status in Appendix A2). In the following, only the six aerosol types (excluding the ice) are considered for comparison between ATLID and MSI aerosol classifications**

**(b) Yes, it is a known issue. The same issue occurred already in the HETEAC paper (Wandinger et al., 2023a), where the aerosol types from the A-PRO processor are discussed as well. We cannot change them in the current paper as it is just using the A-PRO types. By the way, regarding the types, we just use the correct grammatical description. Each aerosol type should be a noun; otherwise you cannot formulate correct sentences. Dust, smoke, and pollution are nouns. Marine and dusty are adjectives and require a noun behind.**

*Because they are many processors and products discussed in the paper, it would be helpful for the reader if the abbreviations don't change during the different sections of the paper. A confusing example is the AM-CTH product which is presented in Section 3.1 and Figure 2 with this name, while later on in Section 4.1.2 it is discussed both as AM-CTH and "CTH detected by AM-COL", with its legend in the plots in fig 7 (and 8,9,10) to be "CTH AM-COL", and in Section 4.1.3 is discussed as "AM-COL CTH" or "CHT AM-COL". It is advised to describe at first from which processor each product is derived and then continue in the presentation of the flowcharts, plots, and discussions mentioning the product name (eg. AM-CTH for this case). Another case is the M-CLD or MIS CTHs in the text (eg. page 17 line 316 and 374) which in the plots is CTH M-CLD and again it would be nice to be homogenized throughout the manuscript.*

**You are right. It is confusing to have the processor and product names mixed throughout the paper. Generally, it is better to use the products and not the processors (see answer to your general comment 1). Therefore, I've changed the figures and the text to state, e.g., CTH AM-CTH and CTH M-COP or AOT at 355 nm (AM-ACD). The CTH is the main product of AM-CTH but for consistency I will state CTH AM-CTH.**

***Specific comments***
***Page 1, line 20:*** *"Two definitions of the CTH from the model-truth cloud extinction fields are compared": if there is a take-home message from this comparison, it would be interesting to be included in the abstract.*

**Thank you. We've added the following statement:**
**Line 22: The geometric CTH is always higher or equal to the radiative CTH.**

***Page 3, line 71:*** *"The dominant aerosol type can be compared to the aerosol mixing ratios applied in M-AOT." This is confusing, as is not clear what is done. Can this be revised to be more clear? Or else add a note for the reader that this will be presented/discussed in section 3.2.1.*

**You are right, the statement is confusing for the reader. We improved the whole paragraph to be more precise.**

**Lines 70-73: The M-AOT algorithm provides aerosol mixing ratios retrieved from MSI observations. The most robust way to compare the ATLID and MSI retrieved aerosol mixing ratios is the comparison of the dominant aerosol type, which is done in the ATLID–MSI Aerosol Column Descriptor (AM-ACD) algorithm.**

***Page 3, line 71:*** *"The combination of ATLID observations at 355 nm with MSI retrievals for wavelengths ≥ 670 nm (Docter et al., 2023) further supports the aerosol typing." Is not very clear what/how this is used. Can you elaborate a little? Even if this will be mentioned in any of the 2 papers referred earlier, 1-2 sentence can be useful to the reader. 2*

**We agree that more details are needed at this point. We were mainly referring to the Ångström exponent. We changed the sentences to the following:**

**Lines 73-75: The Ångström exponent calculated from the ATLID observations at 355 nm and the MSI retrievals at wavelengths ≥ 670 nm (Docter et al., 2023) further constraints the aerosol typing because the spectral behavior contains information about the particle size.**

***Page 4, lines 94+:*** *Could you include the Sentinel 5P CTH retrievals in the 2.1 overview? This would be relevant to the reader who may need to use simultaneously EarthCARE and Sentinel 5P/5 for applications (e.g. for data assimilation).*

**We added a reference to TROPOMI on board the Sentinel-5P and to the future PACE mission (lines 97-99 and 110).**

**Loyola, D. G., Gimeno García, S., Lutz, R., Argyrouli, A., Romahn, F., Spurr, R. J. D., Pedergnana, M., Doicu, A., Molina García, V., and Schüssler, O.: The operational cloud retrieval algorithms from TROPOMI on board Sentinel-5 Precursor, Atmospheric Measurement Techniques, 11, 409–427, https://doi.org/10.5194/amt-11-409-2018, 2018.**

**Compernolle, S., Argyrouli, A., Lutz, R., Sneep, M., Lambert, J.-C., Fjæraa, A. M., Hubert, D., Keppens, A., Loyola, D., O'Connor, E., Romahn, F., Stammes, P., Verhoelst, T., and Wang, P.: Validation of the Sentinel-5 Precursor TROPOMI cloud data with Cloudnet, Aura OMI O2–O2, MODIS, and Suomi-NPP VIIRS, Atmospheric Measurement Techniques, 14, 2451–2476, https://doi.org/10.5194/amt-14-580 2451-2021, 2021.**

**Sayer, A. M., Lelli, L., Cairns, B., van Diedenhoven, B., Ibrahim, A., Knobelspiesse, K. D., Korkin, S., and Werdell, P. J.: The CHROMA cloud-top pressure retrieval algorithm for the Plankton, Aerosol, Cloud, ocean Ecosystem (PACE) satellite mission, Atmospheric Measurement Techniques, 16, 969–996, https://doi.org/10.5194/amt-16-969-2023, 2023.**

***Page 5, line 142:*** *"wind lidar mission Aeolus": It would be nice if you could add a reference here for Aeolus mission or Aladin lidar.*

**We added the following reference (line 146):**
**Stoffelen, A., Pailleux, J., Källén, E., Vaughan, J. M., Isaksen, L., Flamant, P., Wergen, W., Andersson, E., Schyberg, H., Culoma, A., Meynart, R., Endemann, M., and Ingmann, P.: The Atmospheric Dynamics Mission for Global Wind Field Measurement, Bulletin of the American Meteorological Society, 86, 73 – 88, https://doi.org/10.1175/BAMS-86-1-73, 2005.**

***Page 7, lines 162-164:*** *"The A-LAY products … are already provided on JSG with this resolution (approximately 1 km) along track ... The MSI products … are provided on the finer resolution of the MSI grid (500 m)... The surrounding nine MSI pixels correspond to one JSG*

*pixel": With 1 center pixel and 8 surrounding pixels (9 in total) of 500 m JSG would have 1.5 km resolution. How can 9 surrounding pixels of 500 m correspond to 1 km JSG along track? Maybe an explanatory diagram would clarify this question.*

**A good idea. We've added a new figure (Fig. 2a) to illustrate how the MSI grid is mapped to JSG.**

**Figure 2***: It would be very helpful to the reader if the flowchart is more detailed, including not only the steps followed but also the decisions in each step. So one can get from the flowchart all the information for which pixels AM-CTH data are provided and how.*

**We agree. Figure 2 (now Fig. 3) was improved to clearly show which decisions are made by the algorithm. Overlapping lines are eliminated.**

**Page 10, line 227:** *"(default 75 pixels in each direction along track)". Can you include here the distance in km this refers to? In MSI grid, this would mean 37.5 km along the track, in JSG grid of 1 km, this would mean 75 km.*

**We clarified the statement:**

**Lines 231-232: (default 75 JSG pixels (approximately 75 km) in each direction along track)**

**Additionally, we added Fig. 2b to illustrate the process.**

**Page 10, lines 225-229**: *The search for agreement starts at the closest along-track pixel. It continues by searching one pixel before …and one pixel after … from the closest pixel along track … This alternating search is continued until an agreement is found or the configurable maximum search distance … is reached. If a measurement at swath fits to an along-track measurement for all criteria, then the observed CTH difference from the track is assigned to the swath pixel". When reading this part is a little confusing. Only for this one swath pixel the CTH difference is assigned? And then the search for agreement stops for a more far-away grid? Please revise if it is not the case and all pixels are searched until a disagreement is found (which would be the expected case).*

**The search is done for every across-track pixel individually. If an agreement was found, the next across-track pixel is taken, and so on. If no agreement was found, no CTH difference is assigned to the pixel. Neighboring pixel are not considered.**
**In the revised version, Fig. 2b and the decision tree in Fig. 3 are added to make clear how the search is done. In the caption of Fig. 2b it is written:**

**"The sketch illustrates the transfer of the CTH difference from the track to the swath. For an across-track pixel, first the nearest along-track is compared (5 or 3 criteria, see Fig. 3). If no agreement was found, the search continues alternating North (n-1) and South (n+1) of the closest along-track pixel until agreement is found or a configurable maximum search distance is reached. Then, the process is repeated for the next across-track pixel."**

**Figure 3:** *Same suggestion as for figure 2.*

**Figure 3 (now Fig. 4) was redesigned to better show the steps which are made by the AM-ACD algorithm.**

*Page 11, line 243:* "Seven aerosol types (dust, marine aerosol, continental pollution, smoke, dusty smoke, dusty aerosol mix, ice)..." This is very confusing. Why ice is in aerosol types and not in cloud types? Is this the case for the EarthCARE Aerosol type product? Why it couldn't be included in the cloud types, as is the case of CALIPSO?

**At this point, we refer to our answer to your general comment 2. Here a quote from Irbah et al., 2023 (AC-TC paper) is provided:**
**Ice crystals are also treated as an aerosol type, since they span part of the δ–S parameter space not occupied by aerosols (see Fig. 3). This is used to further refine the separation between thin ice clouds from aerosol fields. This is required, since the backscatter-threshold-based cloud aerosol separation step applied earlier tends to produce halos of aerosol around upper-level ice clouds. (Irbah et al., 2023)**

*Page 11, line 243:* "Seven aerosol types (dust, marine aerosol, continental pollution, smoke, dusty smoke, dusty aerosol mix, ice)..." Why marine and dusty mix have in their name additionally the "aerosol" wording, while not all the other aerosol types? CALIPSO has "marine" and dust mixtures types also, without the "aerosol" addition specifically for this type. Can this be harmonized for EarthCARE aerosol types also? Eg "dust, marine, continental pollution, smoke, dusty smoke, dusty mixtures"?

**Please see our answer above to your general comment 2.**

*Page 11, line 243:* "Seven aerosol types (…) are distinguished". Here it would be useful to mention from which processor and in which product the aerosol types are provided.

**Thank you. We added an improved description (see next comment).**

*Page 11, line 244:* "If the aerosol type ice dominates the column integrated aerosol classification, a cirrus cloud is included in the profile which was not detected by the A-CTH algorithm". (a) This is very confusing. If there is an ice cloud, it should be included in the A-CTH product and not be treated from the AM-CTH. And not in the Aerosol types. Why this is not the case? (b) You state that "If the aerosol type ice dominates…". If ice is present but doesn't dominated, is the pixel again excluded? I believe it should be.

**You're right. If the ice contribution is larger than a configurable threshold (default 20% in terms of AOT), a warning is raised. Ice must not be dominating but having a significant contribution is enough to exclude it from the AM-ACD algorithm. However, it is not put into the AM-CTH algorithm because this algorithm runs before AM-ACD. The manuscript was improved to address this and the previous comment:**

**Lines 249-257: Six aerosol types (dust, marine aerosol, continental pollution, smoke, dusty smoke, dusty aerosol mix) and ice are distinguished in the A-TC product (Irbah et al., 2023). The ice is considered to indicate the presence of optically thin ice-containing layers (e.g., diamond dust, subvisible cirrus) that have not been identified as clouds and thus occur in the aerosol products (Irbah et al., 2023; Wandinger et al., 2023b). If the aerosol type ice amounts to a significant contribution (> 20% in terms of AOT, configurable) of the column integrated aerosol classification, a cirrus cloud is included in the profile which was not detected by the A-CTH algorithm. The profile is therefore not cloud free and a warning is raised (see quality status in Appendix A2). In the following, only the six aerosol types (excluding the ice) are considered for comparison between ATLID and MSI aerosol classifications.**

***Page 12, of section 3.2.1 and Table 3:*** *With the description provided on this page for section 3.2.1, it is not clear how the comparison will reach agreement or not. Can Table 3 be enhanced with the used thresholds of the agreement for each A-TC type? Also, can one column with the M-AOT aerosol classification be included in the Table?*

**The M-AOT aerosol classification uses the 4 aerosol components of HETEAC. In Docter et al., 2023, a table defines the 25 precalculated mixtures of the 4 HETEAC aerosol components. If one of 4 basic aerosol components dominates the mixture in the M-AOT product and the corresponding A-TC aerosol type dominates the columnar aerosol classification probabilities in the A-ALD product, agreement is reached. Here, no thresholds are necessary. The thresholds get necessary when it comes to the mixed aerosol types in A-TC.**
**We tried to make it clearer throughout the subsection by pointing to M-AOT. Another column in the table seems not to be necessary, as M-AOT uses exactly the four basic aerosol components defined in HETEAC. The comparison would have been easier, if A-TC would have used the 4 HETEAC aerosol components instead of the 6 aerosol types which are mixtures of the HETEAC aerosol components.**

***Page 13, line 268:*** *"If the dominant aerosol type agrees (see Sect. 3.2.1)". It would be helpful in this section to mention how the dominant aerosol type is defined. Eg., the M-AOT HETEAC component with the biggest %?*
**Thank you for pointing to this issue. We improved the code in the way, that the Ångström exponent is calculated for the 6 (A-TC) aerosol types along track. Across track, the dominant aerosol type is determined from the 4 HETEAC aerosol components according to the description in Sect. 3.2.1 (4 pure types are mapped directly, for the mixtures the thresholds are applied. We added a definition of the dominant aerosol type:**

**Line 264: The dominant aerosol type is defined by the highest columnar aerosol classification probability (A-ALD product).**

***Figure 6****: How y axis density is calculated? Scaled to the total number of pixels for every case, with 1 as a cumulative sum? Maybe is worth mentioning it. Also, the colorbar in model truth comparison plot (and relevant plots from there on) can use a legend/units (eg. nr pixels).*

**Thank you for your comment. Indeed, we revised the legend and changed the plotting from density (where it is normalized to sum of all pixel) to number of pixels. Now, it get's clear that AM-CTH provides values for approximately half of the cloudy pixels detected by M-CM. Therefore, the number of counts in brackets was removed. A caption was also given to the color bar in the scatter plots.**

***Page 17, lines 375-376****: "Especially the cirrus clouds between 9 and 13 km height are detected by AM-CTH above a COT of 0.25". This is confusing. From Figure 6 I would conclude that the cirrus clouds between 9 and 13 km height are detected by AM-CTH below a COT of 0.25. But maybe there is something else you wanted to highlight. Please rephrase to make it clear.*

**Sorry, thank you for pointing to it. It should be written "below a COT of 0.25". We changed it.**

***Page 18, line 378****: "The amount of data points within an interval of ±i m around the 1:1 line (fi in Fig. 7 and 8) shows a similar behavior for AM-COL to extinction-based model truth (40, 63, 83% for 300, 600, 1500 m) and M-CLD to COT-based model truth (31, 53, 77% for 300, 600, 1500 m)". Earlier in the manuscript (page 17 line 366) was mentioned that "40% are within ±300 m which was defined in the mission requirements". Does the statistics on page 18 show*

*us that only the AM-COL is within the mission requirements, while the M-CLD isn't? Please consider if you would like to highlight it in this part of the paper.*

**The aim of this statement is to show that AM-CTH provides you the geometric CTH obtained by the extinction threshold and M-COP provides the radiative CTH as obtained by the COT threshold. At this point the mission requirements are not accurately enough as they state just CTH. Nevertheless, I would prefer the geometric definition when talking about CTH. Therefore, AM-CTH better fulfills the mission requirements. However, M-COP is needed to provide information besides the track. Therefore, we do not want to emphasize that just one processor is within the mission requirements. The combination of both instruments brings the best results – along with the synergistic approach of EarthCARE. We have not changed the text at this point.**

*Page 22, line 418: "Thus, the dominant". Why thus? Could it be the case that the classifications are not so successful, hence "thus" is not correct? Or there is a connection between the simplicity of the scene and the fact that the classifications are successful? If possible, modify the text to make it clear.*

**Thank you. We deleted the word "Thus".**

*Page 22, lines 420-422: "The ice cloud at 34°N was only partly detected by the MSI cloud mask and thus the AOT of the ice crystals is included in the M-AOT product". One wouldn't expect to find ice OD in AOT products. Why this is not the case for EarthCARE products?*

**The reason is simply that the pixels were marked as cloud-free by M-CM and therefore the aerosol processor M-AOT was applied to them. The ice crystals contribute to the optical depth and M-AOT retrieves the optical depth for the pixels which were classified as cloud-free. But you're right the word AOT is not correct in this sentence. We've changed it to:**

**Lines 448-449: The ice cloud at 34°N is only partly detected by the MSI cloud mask and thus the optical thickness of the ice crystals is included in the M-AOT product.**

*Page 22, line 420-422: "Here, as well the ice crystals are included in the AOT, which differs from the CAMS model truth AOT provided for aerosol only". Please revise to improve the syntaxis.*
**Thank you. We revised the statement:**

**Lines 450-452: The additional optical thickness of the ice crystals increases the AOT in the A-ALD product and lead to an overestimation compared to the CAMS model truth AOT which is provided for aerosol only.**

*Figure 12: Can you comment on why some values (with the highest AOT) are flagged out in the 355 nm AOT, although some seem to have quality status = 0?*

**Thank you for your careful look. Indeed, the quality status was not well defined. We've improved the algorithm and the manuscript. Now, the quality status for AM-ACD is defined as:**

**QACD = 0:    Good data, high quality of M-AOT input.**

**QACD = 1:    Warning: A significant amount of ice (> 20% (configurable) in terms of AOT) was detected by A-TC (provided in A-ALD). This warning is provided along track only, but probably holds for the close swath pixel as well.**

**QACD = 2:** Warning: Dominant aerosol type on swath was not present along the track, AOT at 355 nm could not be calculated.

**QACD = 3:** Warning: The homogeneity criteria of M-AOT are not fulfilled.

**QACD = 4:** Bad data. Observations on MSI grid are not consistent on JSG.

**QACD = −1:** Not surely cloud free according to M-CM.

And Fig. 12 (now Fig. 14) was updated accordingly.

*Page 22-23, lines 434-439: "The derived …at 607 nm". Is there an error estimation for this new product (AM-ACD AOT 355)? If yes, does it consider/include the uncertainty due to the Ångström exponent bias mentioned?*

Yes, there is an error estimation of the AM-ACD AOT at 355 nm. The uncertainty of the Ångström exponent was included as an additional source of uncertainty for the product. However, the bias of the Ångström exponent is only known for the simulated test scenes. It won't be the case for real world data.

*Page 35, lines 469-470: "However, the brightness temperature difference between 10.8 and 12.0 μm was not sensitively enough simulated to clearly detect multi-layer cloud scenarios by MSI." I believe that the brightness temperature sensitivity is not discussed earlier when the results from the multi-layer cloud scenarios are presented. It would be interesting to include a comment on this in the earlier session also.*

A further sentence was added in the subsection 4.1.3 (AM-CTH algorithm performance for different cloud classes) when the multilayer cloud detection was described.

Lines 408-410: However, the brightness temperature difference between 10.8 and 12.0 μm was not sensitively enough simulated in the EarthCARE test scenes to clearly detect multi-layer clouds with MSI.

*Page 27: "QCTH = 4: Bad data. Observations on MSI grid are not consistent on (coarser) JSG". Coarser JSG is not defined in the text. Can you define it here?*

The resolution of the Joint Standard Grid (JSG) is coarser than the one of the MSI grid. However, there is only one JSG and no "coarser JSG". We deleted the word "coarser".

*Technical corrections/suggestions (bold text & red ",:"):*

Your careful reading is highly appreciated. We followed your suggestions in the revised version.

*Page 1, line 1: "is **a** combination of multiple active…".*
*Page 1, line 6: "characterize the 3-dimensional scene", a suggestion to change to "characterize **a** 3-dimensional scene", or "characterize the 3-dimensional scene**s**".*
*Page 1, line 7: "(A-LAY)*,  *MSI L2a data from the MSI Cloud Products processor (M-CLD),*  **[here we kept the 'and']** *the MSI Aerosol Optical Thickness processor (M-AOT)*, *as well as MSI Level 1c (L1c) data are used as input to produce the synergistic columnar products".*
*Page 1, line 14: "CTH detected by ATLID **and retrieved/provided** MSI is calculated": retrieved or provided by MSI is a more representative term for this product.*

*Page 1, line 18: "The quality status depending on day/night conditions or the presence of multiple cloud or aerosol layers is provided with the products". The syntax could be improved.*

*Page 2, line 34: "**a** three-dimensional (3D) scene**s** (e.g., Qu et al., 2022a; Mason et al., 2022) to calculate  radiative flux**es** which  **are** compared..".*

*Page 2, line 40: "about the scene  **around** the satellite track".*

*Page 3, line 46-48: "It provides vertical profiles along the satellite track of the particle backscatter and extinction coefficient, the lidar ratio**,** and the particle linear depolarization ratio which are  **stored** in the ATLID L2a product A-EBD". Suggestion because they are 2 provided in 1 sentence and is less clear.*

*Page 3, line 52-53: "et al., 2022)**,** and to retrieve cloud optical properties such as the cloud optical thickness (COT), CTH and the effective radius of the cloud droplets which  **are** provided in the MSI Cloud Optical and Physical product".*

*Page 3, line 56: "for  **a** 3D scene". Is it a requirement for this scene presented in the paper, or for overall the scenes? If the 2nd "a" is needed there.*

*Page 3, line 9:"**a** reasonable estimate**s**".*

*Page 4, line 84: " using  common test scenes".*

*Page 4, line 86: "Conclusion**s**"*

*Page 4, line 96: "with lidar**s**  **example** from the Cloud-Aerosol Lidar and Infrared Pathfinder Satellite Observations": CALIPSO is not the only lidar that has been used for CTH detection.*

*Page 4, line 100: "in dependence **on**  the type".*

*Page 4, line 105: "the CTH  **of** high".*

*Page 5, line 119: "not from  **a** space lidar".*

*Table 1: "and the products () in which they are contained **(bold, with references)**".*

*Page 7, line 158: "…after the  ATLID L2a and MSI L2a processing is completed".*

*Page 8, line 180: " **Then** the scene…".*

*Page 10, line 234: "over ocean)**, and** the respective Ångström exponents, and their uncertainties".*

*Page 14, lines 299-300: "**And more specifically w**With the EarthCARE End-to-End Simulator specific test scenes **which** were created to test the full chain of EarthCARE processors". Something is missing in this sentence. A possible suggestion.*

*Page 14, line 327: "There are several reasons:": It would read better if you clarify after this text the reasons. Eg. "reasons of failure" "reasons AM-CTH can't be retrieved".*

[revised manuscript text omitted]